# Technical note: Sensitivity of the CAMS regional air quality modelling system to anthropogenic emission temporal variability

Marc Guevara<sup>1</sup>, Augustin Colette<sup>2</sup>, Antoine Guion<sup>2</sup>, Valentin Petiot<sup>3</sup>, Mario Adani<sup>4</sup>, Joaquim Arteta<sup>3</sup>, Anna Benedictow<sup>5</sup>, Robert Bergström<sup>6</sup>, Andrea Bolignano<sup>4</sup>, Paula Camps<sup>1</sup>, Ana C. Carvalho<sup>6</sup>, Jesper Heile Christensen<sup>7</sup>, Florian Couvidat<sup>2</sup>, Ilia D'Elia<sup>4</sup>, Hugo Denier van der Gon<sup>8</sup>, Gaël Descombes<sup>2</sup>, John Douros<sup>9</sup>, Hilde Fagerli<sup>5</sup>, Yalda Fatahi<sup>10</sup>, Elmar Friese<sup>11</sup>, Lise Frohn<sup>7</sup>, Michael Gauss<sup>5</sup>, Camilla Geels<sup>7</sup>, Risto Hänninen<sup>10</sup>, Kaj Hansen<sup>7</sup>, Oriol Jorba<sup>1</sup>, Jacek W. Kaminski<sup>12</sup>, Rostislav Kouznetsov<sup>10</sup>, Richard Kranenburg<sup>8</sup>, Jeroen Kuenen<sup>8</sup>, Victor Lannuque<sup>2</sup>, Frédérik Meleux<sup>2</sup>, Agnes Nyíri<sup>5</sup>, Yuliia Palamarchuk<sup>10</sup>, Carlos Pérez García-Pando<sup>1,13</sup>, Lennard Robertson<sup>6</sup>, Felicita Russo<sup>4</sup>, Arjo Segers<sup>8</sup>, Mikhail Sofiev<sup>10</sup>, Joanna Struzewska<sup>12</sup>, Renske Timmermans<sup>8</sup>, Andreas Uppstu<sup>10</sup>, Alvaro Valdebenito<sup>5</sup>, Zhuvun Ye<sup>7</sup>

Correspondence to: Marc Guevara (marc.guevara@bsc.es)

Abstract. An accurate characterization of the temporal distribution in primary emissions is essential for air quality modeling. This study evaluates the impact of replacing the default temporal profiles in the Copernicus Atmosphere Monitoring Service (CAMS) European air quality multi-model ensemble with an updated dataset (CAMS-REG-TEMPO). The sensitivity of 11 regional models and the ensemble to these changes is assessed by comparing modeled and observed monthly, weekly, and diurnal cycles of nitrogen dioxide (NO<sub>2</sub>), ozone (O<sub>3</sub>), coarse particulate matter (PM<sub>10</sub>), and fine particulate matter (PM<sub>2.5</sub>) across Europe. NO<sub>2</sub> shows the greatest improvement, with weekly cycle correlations increasing up to +0.17 due to better road transport emissions representation. PM<sub>10</sub> correlations improve in winter (up to +0.13 weekly and +0.07 diurnal) due to refined residential wood combustion emissions. PM<sub>2.5</sub> correlations remain largely unchanged, except for diurnal cycles, which improve in winter (+0.18) but slightly degrade in spring and summer (-0.02). O<sub>3</sub> is the least affected, as correlations were already high with default profiles (0.9–0.95). For some species and timescales (e.g., NO<sub>2</sub> diurnal cycles), results vary across models,

<sup>&</sup>lt;sup>1</sup> Barcelona Supercomputing Center, Barcelona, 08034, Spain

<sup>&</sup>lt;sup>2</sup> Institut National de l'Environnement Industriel et des Risques, Verneuil en Halatte, 60550, France

<sup>&</sup>lt;sup>3</sup> Météo-France, Saint-Mandé, 94165, France

<sup>4</sup> ENEA: Italian National Agency for New Technologies, Energy and Sustainable Economic Development, Bologna, 40129, Italy

<sup>&</sup>lt;sup>5</sup> MET Norway: Norwegian Meteorological Institute, Oslo, 0372, Norway

<sup>&</sup>lt;sup>6</sup> SMHI: Swedish Meteorological and Hydrological Institute. Norrköping, SE-601 76, Sweden

<sup>&</sup>lt;sup>7</sup> Aarhus University: Roskilde, 4000, Denmark

<sup>&</sup>lt;sup>8</sup> TNO: Netherlands Organisation for applied scientific research, Utrecht, 3584, The Netherlands

<sup>&</sup>lt;sup>9</sup> KNMI: Royal Netherlands Meteorological Institute, De Bilt, 3730, The Netherlands

<sup>&</sup>lt;sup>10</sup> FMI, Finnish Meteorological Institute, Helsinki, 00-001, Finland

<sup>&</sup>lt;sup>11</sup> Forschungszentrum Jülich GmbH, ICE-3, Institute of Climate and Energy Systems - Troposphere, 52428 Jülich, Germany

<sup>&</sup>lt;sup>12</sup> IEP-NRI: Institute of Environmental Protection - National Research Institute, Warsaw, 00-001, Poland

<sup>25 &</sup>lt;sup>13</sup> Catalan Institution for Research and Advanced Studies (ICREA), 08010, Barcelona, Spain

highlighting the complex interactions between emission timing and atmospheric processes. CAMS-REG-TEMPO has little effect on annual RMSE and bias, aside from slight improvements in high PM<sub>10</sub> concentrations. Overall, the findings support implementing CAMS-REG-TEMPO in the operational CAMS multi-model ensemble.

### 1 Introduction

50

Air quality models require hourly emissions from primary pollutants to accurately represent dispersion and physico-chemical processes in the atmosphere. Numerous studies have demonstrated that a precise temporal distribution of emissions is crucial for capturing observed patterns from both ground-based and satellite observations (e.g., Mues et al., 2014; Fatahi et al., 2021, Skjøth et al. 2011, Baek et al., 2023; Grythe et al., 2019; Super et al., 2021). Despite the critical role of temporally resolved emissions on model performance, there are currently no international regulations mandating the reporting of emission inventories at such fine level of temporal disaggregation. As a result, emission inventories used for air quality modelling activities are typically provided at the annual or monthly levels. To achieve the necessary temporal granularity, emissions must be downscaled using predefined temporal weight factors at different levels: month-of-the-year (i.e., monthly), day-of-the-week (i.e., weekly) and hour-of-the-day (i.e., hourly) temporal weight factors.

At the European level, emission temporal profiles developed or derived from studies conducted in the late 1990s and early 2000's (e.g., Ebel et al., 1997) are still being widely used by multiple air quality modelling teams. This includes the European regional air quality production service provided by the Copernicus Atmosphere Monitoring Service (CAMS), which operationally delivers air quality daily analyses, forecasts and reanalyses through a multi-model ensemble approach (Colette et al., 2024). However, recent studies have identified limitations in these profiles, such as the reliance on outdated sources of information and failure to account for sociodemographic influences and climatological conditions (e.g., Backes et al., 2016a; Athanasopoulou et al., 2017). Moreover, the recently revised Ambient Air Quality Directive 2024/2881/EC in Europe set more stringent standards to be attained by 2030, acknowledging modelling applications as a fundamental support in the assessment of air pollution. CAMS delivers operational products suited and designed for supporting the implementation of the AAQD, which pushes for continuous improvement of current products accuracy (e.g., de Meij et al., 2025). To overcome these challenges and improve the representation of temporal variations in emissions used for modelling applications, a new dataset of temporal profiles —CAMS-TEMPO— was recently developed within the CAMS framework (Guevara et al., 2021).

The aim of this study is to analyse and quantify the impact of implementing the new CAMS-TEMPO anthropogenic temporal profiles on the performance of the CAMS multi-model ensemble. The sensitivity of the 11 regional models that comprise the CAMS ensemble is assessed by comparing modelling results against observations from a European network of air quality ground-based stations. The analysis shows how changes in emission temporal distribution affect the ability to reproduce observed monthly, weekly and diurnal cycles of four key air pollutants: nitrogen dioxide (NO<sub>2</sub>), ozone (O<sub>3</sub>), particulate matter

(PM<sub>10</sub>) and fine particulate matter (PM<sub>2.5</sub>). Changes in the average deviation from observations are also analysed. A key contribution of this study, compared to previous research on emission temporal variations (e.g., Mues et al., 2014), is its comprehensive evaluation across multiple models. By drawing conclusions from a diverse ensemble rather than individual models, this approach minimizes the risk of error compensation and provides a more robust assessment of emission temporal effects on air quality modelling. Testing the impact of changing emission temporal profile with a single model carries a risk to correct a bias which would be actually due to the misrepresentation of other factors affecting the daily or seasonal variability (typically planetary boundary layer or insolation variability). While we cannot rule out that such misrepresentation occur in several models, it is relatively unlikely that it would act in the same direction in the whole ensemble. That is why the ensemble approach mobilised here argues in favour of the robustness of the diagnostic.

The methods and data used in this work are presented in Sect. 2. The results section (Sect. 3) discusses the temporal distribution analysis for primary emissions, and the temporal correlation analysis and the mean deviation analysis for modelled air pollutant concentrations. Finally, Sect. 4 summarises the main conclusions and lessons learned.

# 2 Method and data


# 2.1 The CAMS regional air quality modelling system

The CAMS regional service (https://atmosphere.copernicus.eu/european-air-quality-forecast-plots/) provides daily 4-day forecasts for key air quality species along with analyses of the previous day, and retrospective reanalyses using the latest observation datasets available for assimilation. As the reference air quality forecasting system at the European scale, it operates through a distributed network of eleven Chemical Transport Models (CTMs) across ten European countries (described in Table 1), coordinated by a Centralised Regional Production Unit to ensure consistency. Using an ensemble of CTMs enhances forecast reliability by reducing the risk of failure in daily production and improving the skill of the forecast (Galmarini et al., 2013). Detailed information on the CAMS regional air quality production system and the individual models within the ensemble can be found in Colette et al. (2024). While each model differs in its design with regards to internal physical and chemical processes, strong common requirements exist in the CAMS regional service with regards to forcing meteorological data, chemical boundary conditions at the European boundary, and anthropogenic emissions.

Table 1 Chemistry transport models participating in the CAMS regional ensemble system

| Model name | Institute / country                                                          | Reference           |  |  |
|------------|------------------------------------------------------------------------------|---------------------|--|--|
| CHIMERE    | Institut National De L'environnement<br>Industriel Et Des Risques (INERIS) / | Menut et al. (2021) |  |  |
|            | France                                                                       |                     |  |  |

| Danish Eulerian<br>Hemispheric Model<br>(DEHM)                                                 | Aarhus University / Denmark                                                                                                                            | Christensen (1997), Brandt et al. (2012), Geels et al. (2021), Frohn et al. (2002 and 2021)  |
|------------------------------------------------------------------------------------------------|--------------------------------------------------------------------------------------------------------------------------------------------------------|----------------------------------------------------------------------------------------------|
| European Monitoring<br>and Evaluation<br>Programme (EMEP)                                      | Norwegian Meteorological Institute – (MET Norway) / Norway                                                                                             | Simpson et al. (2012), EMEP MSC-W (2022)                                                     |
| European Air<br>pollution Dispersion -<br>Inverse Model<br>(EURAD-IM)                          | Forschungszentrum Jülich Institute of<br>Climate and Energy Systems<br>Troposphere (FZJ ICE-3) / Germany                                               | Franke et al. (2024), Friese and Ebel (2010)                                                 |
| Global Environmental Multiscale model - Air Quality chemistry (GEM-AQ)                         | Institute of Environmental Protection  – National Research Institute (IEP-<br>NRI) / Poland                                                            | Kaminski et al. (2008), Struzewska and Kaminski (2008)                                       |
| Long Term Ozone<br>Simulation –<br>European Operational<br>Smog model<br>(LOTOS-EUROS)         | Royal Netherlands Meteorological<br>Institute (KNMI) and The<br>Netherlands Organisation for Applied<br>Scientific Research (TNO) / The<br>Netherlands | Manders et al. (2017)                                                                        |
| Multi-scale Atmospheric Transport and Chemistry model (MATCH)                                  | Swedish Meteorological and<br>Hydrological Institute (SMHI) -<br>Sweden                                                                                | Roberston et al. (1999); Andersson et al. (2007)                                             |
| National Integrated Model to support International Negotiation on Air Pollution issues (MINNI) | Italian National Agency for New<br>Technologies, Energy and<br>Sustainable Economic Development<br>(ENEA) / Italy                                      | D'Elia et al. (2021); Mircea et al. (2014)                                                   |
| Modèle de Chimie<br>Atmosphérique de<br>Grande Echelle<br>(MOCAGE)                             | Météo-France / France                                                                                                                                  | Josse et al. (2004), Sič et al. (2015); Guth et al. (2016)                                   |
| Multiscale Online Nonhydrostatic AtmospheRe CHemistry model (MONARCH)                          | Barcerlona Supercomputing Center<br>(BSC) / Spain                                                                                                      | Badia et al. (2017), Klose et al. (2021), Navarro-Barboza et al. (2024), Pérez et al. (2011) |
| System for Integrated Modeling of Atmospheric Composition (SILAM)                              | Finnish Meteorological Institute<br>(FMI) / Finland                                                                                                    | Sofiev et al. (2015, 2010 and 2018), Sofiev (2002),<br>Kouznetsov and Sofiev (2012)          |

#### 2.2 Emission inputs






The CAMS European regional air pollutant emission inventory (CAMS-REG-AP\_v4.2; Kuenen et al., 2022) is used to represent anthropogenic emissions. This inventory uses official annual air pollutant emissions submitted by each country to the European Monitoring and Evaluation Programme (EMEP) and performs a spatial mapping to a grid of 0.1×0.05 degrees using appropriate surrogate statistics for each activity. Some examples of spatial proxies include a road transport network with traffic intensities associated to each road link, which is used to distribute interurban traffic emissions, and a catalogue of industrial point sources with exact geographical coordinates and emission strengths associated to each facility, which are used to distribute emissions from power plants and manufacturing industries. The summary of proxies used is provided in Kuenen et al. (2022). NMVOC and PM emissions are speciated using the sector- and country-dependent speciation profiles provided in CAMS-REG, which allow break downing the total NMVOC to the 25 Global Emission InitiAtive (GEIA) species (Schultz et al., 2007) and the total PM emissions to primary organic carbon, elemental carbon, sulphates, sodium and others. Each individual CAMS modelling team performs a remapping of the 25 GEIA NMVOC species and individual PM component to the species used in their corresponding gas phase and aerosol chemical mechanisms. Biomass burning emissions are derived from the CAMS Global Fire Assimilation System (GFAS; Kaiser et al., 2012) across all CAMS regional models, while emissions from other natural sources such as biogenic, sea salt and desert dust are estimated by each model system using dedicated and diverse on-line parametrisations, as detailed in the references summarised in Table 1.

# 2.3 Anthropogenic temporal profiles

# 115 **2.3.1 Default profiles**

Table 2 summarises the default temporal profiles used by each model in the CAMS regional production service at the time of performing this study. Most of the models (7 out of 11) perform the temporal disaggregation of the anthropogenic emissions using the profiles constructed by the Netherlands Organisation for Applied Scientific Research (TNO; Denier van der Gon et al., 2011), while the remaining models use the temporal factors from the Generation of European Emission Data for Episodes project (GENEMIS) (Ebel et al., 1997; Friedrich and Reis, 2004). Both datasets were developed at European level and include monthly, weekly and diurnal temporal profiles.

GENEMIS monthly and weekly profiles vary per sector and country, while hourly profiles vary per sector only. The profiles were determined using various indicators, including fuel use, power plant load curves, temperature, heating degree days, working hours, traffic counts and fertilizer use, among others (Lenhart and Friedrich, 1996). In contrast, TNO profiles are sector-dependent only across all timescales (monthly, weekly and hourly) and largely based on GENEMIS data and older Western European datasets. For example, road transport profiles are based on Dutch traffic count data from 1985-1998, while energy sector profiles are derived from power plant fuel usage and load curves reported by Veldt (1992). Livestock emissions in TNO profiles are based on Skjøth et al. (2011), which developed a dynamic emission model that takes into account the

effect of outdoor temperatures in NH<sub>3</sub> emissions from animal houses or manure storages. Both GENEMIS and TNO report the same hourly sector-dependent profiles. In the CHIMERE and EMEP models, the GENEMIS hourly weight factors for road transport are replaced by country- and day-of-the-week-dependent profiles developed by Menut et al. (2012), which were derived from measured surface NO<sub>2</sub> concentrations at European traffic stations. More details on the proxies and sources of information considered to construct the TNO and GENEMIS profiles are provided as part of the emission result analysis in 135 Sect. 3.1.1 to 3.1.3.

Table 2 Summary of the default emission temporal profiles used by the CAMS regional models

| Dataset Monthly profil                                                                                                                                                                     |                               | Weekly profiles Hourly profiles      |  | Models                                                              |  |  |  |
|--------------------------------------------------------------------------------------------------------------------------------------------------------------------------------------------|-------------------------------|--------------------------------------|--|---------------------------------------------------------------------|--|--|--|
| TNO profiles                                                                                                                                                                               | Sector-dependent              | Sector-dependent Sector-dependent(1) |  | EURAD-IM, GEM-AQ, LOTOS-<br>EUROS, MINNI, MOCAGE,<br>MONARCH, SILAM |  |  |  |
| GENEMIS profiles                                                                                                                                                                           | Sector- and country-dependent | Sector- and country-dependent        |  | CHIMERE <sup>(2)</sup> , DEHM, MATCH,<br>EMEP <sup>(2)</sup>        |  |  |  |
| (1) The hourly sector-dependent profiles in GENEMIS and TNO are identical (2) Hourly factors for road transport from Menut et al. (2012), which are country, and day-of-the-week-dependent |                               |                                      |  |                                                                     |  |  |  |

#### 2.3.2 **CAMS-REG-TEMPO** profiles

The CAMS REGional TEMPOral (CAMS-REG-TEMPO) dataset consists of a collection of European regional temporal factors aligned with the domain specifications (resolution and geographical coverage) and sector classification of the CAMS-REG-AP emission inventory. It includes monthly, weekly, daily (day-of-the-year) and hourly temporal profiles for the key air pollutants, namely: nitrogen oxides (NO<sub>x</sub>), sulphur oxides (SO<sub>x</sub>), non-methane volatile organic compounds (NMVOCs), ammonia (NH<sub>3</sub>), carbon monoxide (CO), coarse particulate matter (PM<sub>10</sub>) and fine particulate matter (PM<sub>2.5</sub>). Temporal 145 profiles vary in spatial representation depending on the pollutant source and temporal resolution (i.e., monthly, weekly, daily, hourly): some are spatially invariant (i.e., a unique set of temporal weights for the entire domain), while others are spatially variant (i.e., temporal weights vary by grid cell or country). Additionally, profiles may be year-dependent and/or pollutantdependent, depending on the characteristics of the input data and the approaches to compute the profiles. The dataset is built using a wide range of data sources –including energy statistics and measured activity data, among others– and meteorology-150 dependent parametrizations such as the heating degree day approach. A detailed description of the datasets and parametrizations is available in Guevara et al. (2021).

This study considers an updated version of the CAMS-REG-TEMPO dataset first presented in Guevara et al. (2021). The key updates in this new version (v3.2) compared to the previous release (v2.1) are as follows:

- Road transport (GNFR\_F): Updated monthly temporal profiles for urban and rural areas were developed to distinguish between urban and interurban road traffic activities. Urban profiles were derived from TomTom congestion statistics for European cities (<a href="https://www.tomtom.com/en\_gb/traffic-index/">https://www.tomtom.com/en\_gb/traffic-index/</a>). These city level profiles were aggregated to the country level based on the annual average congestion and city population. Rural profiles were constructed using a wide range of traffic count datasets from national road administrations (Table S1). The classification of urban and rural areas within the CAMS-REG-AP grid follows the Global Human Settlement Layer (GHSL) dataset (Pesaresi et al., 2019). For that, the original 1km × 1km GHSL raster was remapped onto the CAMS-REG-AP grid (0.1° × 0.05°) following a majority resampling method, in which each destination grid cell was assigned with the GHSL classification that had a higher number of occurrences within that grid cell. New weekly and hourly temporal profiles were also constructed using TomTom congestion statistics, but without differentiating between urban and rural areas.
  - Aviation (GNFR\_H): In v2.1, a flat (i.e., no variation across time steps) weekly profile was assumed for this sector.
    In v3.2, country-dependent weekly profiles were introduced, derived from daily air traffic statistics at national airports
    from 2016 to 2019 provided by EUROCONTROL (2020). These profiles were aggregated at the country level based
    on the available national airport data.
- Shipping (GNFR\_G): Previously, no monthly variations were considered for this sector. In v3.2, sea region- and pollutant-dependent monthly profiles were developed using CAMS-GLOB-SHIP\_v2.1 AIS-based monthly emissions (Jalkanen et al., 2016). The new profiles vary per pollutant and sea region but are considered yearly independent due to minimal year-to-year variations.
- Other mobile sources (GNFR I): In v2.1, flat monthly, weekly and hourly profiles were assumed for this sector. In 175 v3.2, pollutant-dependent monthly, weekly and hourly profiles were developed using the profiles reported in the EMEP/EEA emission inventory guidebook (EMEP/EEA, 2019) MapEIre and the project (https://projects.au.dk/mapeire/). The profiles reported by EMEP/EEA (2019) include temporal weight factors for Agriculture and Forestry, Industry and Construction, Household and Gardening and Military subcategories. The weight factors for the Commercial and Institutional subcategory were derived from MapEire as they are not included 180 in EMEP/EEA (2019). Subcategory profiles were averaged at the GNFR I level based on their contributions to total GNFR I emissions, estimated considering the 2018 EMEP official reported emission data for the EU27 plus UK (EMEP/CEIP, 2021).
- Data gap-filling procedure: In version 2.1, TNO profiles were applied by default in countries where local proxies (e.g., electricity production, air traffic statistics) were unavailable. In the v 3.2, a more refined approach was adopted by constructing averaged profiles from countries within the same world region, based on world region definitions from the EDGAR emission inventory (Crippa et al., 2018).

Table 3 summarizes the characteristics of each profile included in the CAMS-REG-TEMPO dataset for each sector and temporal resolution. For the fugitive fossil fuel (GNFR\_D), use of solvents (GNFR\_E) and waste management (GNFR\_J)

sectors, as the profiles remain unchanged from those reported by TNO due to lack of more detailed information. However, it is important to highlight that these sectors contribute minimally to total primary European emissions for all pollutants. An exception is GNFR\_E (solvent use), which accounts for approximately 35% of total NMVOC at the EU27 level (EMEP/CEIP, 2021).

To facilitate the integration of the CAMS-REG-TEMPO profiles into CAMS regional models, the gridded profiles were simplified to a country-level format. This process involved combining the original CAMS-REG-TEMPO gridded profiles with the CAMS-REG-AP\_v4.2 annual inventory to generate gridded monthly and daily emissions using the HERMESv3\_GR emission processing system (Guevara et al., 2019). The resulting monthly and daily gridded emissions were then averaged at the country level and normalized to produce country- and pollutant-dependent simplified profiles. This simplification was applied to all emission temporal profiles provided at the grid cell level (Table 3), including monthly profiles for the GNFR\_F sector (all species) and daily temporal profiles for the GNFR\_C sector (all species), GNFR\_K (livestock emissions, NH<sub>3</sub> and NO<sub>x</sub>) and GNFR L (other agricultural emissions, NH<sub>3</sub>).

Table 3 Main characteristics of the CAMS-REG-TEMPO dataset. Per country: indicates that the profiles vary per country; per pollutant: indicates that the profiles vary per pollutant; per grid cell: indicates that the profiles vary per grid cell within a country; per year: indicates that the profiles vary per year; fixed: indicates that the profiles are spatially invariant. The symbol "-" denotes that no profile is proposed.

| Sector                             | Description                                              | <b>Description</b> Monthly                                           |                        | Weekly                                    | Hourly                                              |
|------------------------------------|----------------------------------------------------------|----------------------------------------------------------------------|------------------------|-------------------------------------------|-----------------------------------------------------|
| GNFR_A                             | Public Power                                             | per country, pollutant                                               | -                      | per country, pollutant                    | per country, pollutant                              |
| GNFR_B                             | Industry                                                 | per country                                                          | -                      | fixed (1)                                 | fixed (1)                                           |
| GNFR_C Other stationary combustion |                                                          |                                                                      | per grid cell,<br>year |                                           | per pollutant                                       |
| GNFR_D                             | Fugitive fossil fuel                                     | fixed (1)                                                            | -                      | fixed (1)                                 | fixed (1)                                           |
| GNFR_E                             | Solvents                                                 | fixed (1)                                                            | -                      | fixed (1)                                 | fixed (1)                                           |
| GNFR_F1                            | Road transport exhaust gasoline                          | per year, grid cell for CO<br>and NMVOC; per grid cell<br>for others | -                      | per country                               | per country, day type <sup>(2)</sup>                |
| GNFR_2                             | Road transport exhaust diesel                            | per year, grid cell for NOx;<br>per grid cell for others             | -                      | per country                               | per country, day<br>type                            |
| GNFR_F3                            | Road transport exhaust LPG                               | per grid cell                                                        | -                      | per country                               | per country, day type                               |
| GNFR_F4                            | Road transport non-<br>exhaust (wear and<br>evaporative) | per grid cell for PM; fixed for NMVOC                                | -                      | per country for<br>PM; fixed for<br>NMVOC | per country, day<br>type for PM; fixed<br>for NMVOC |
| GNFR_G                             | Shipping                                                 | per sea region and pollutant                                         | -                      | fixed (1)                                 | fixed (1)                                           |
| GNFR_H                             | Aviation                                                 | per country                                                          | -                      | per country                               | fixed                                               |

| GNFR_I | Off road transport                                    | fixed, per pollutant                        | fixed, per<br>pollutant                                           |                                                                                | fixed, per pollutant |
|--------|-------------------------------------------------------|---------------------------------------------|-------------------------------------------------------------------|--------------------------------------------------------------------------------|----------------------|
| GNFR_J | Waste management                                      | fixed (1)                                   | -                                                                 | fixed (1)                                                                      | fixed (1)            |
| GNFR_K | Agriculture (livestock)                               | fixed for others than $NH_3$ and $NO_x$     | per grid cell,<br>year for NH <sub>3</sub><br>and NO <sub>x</sub> | fixed for others<br>than NH <sub>3</sub> and<br>NO <sub>x</sub> <sup>(1)</sup> | fixed <sup>(1)</sup> |
| GNFR_L | Agriculture (fertilizers, agricultural waste burning) | per country for others than NH <sub>3</sub> | per grid cell,<br>year for NH <sub>3</sub>                        | (fixed for others than NH <sub>3</sub> ) <sup>(2)</sup>                        | fixed, per pollutant |

<sup>(1)</sup> Same profile as the one reported by the TNO dataset (Denier van der Gon et al., 2011)

#### 2.4 Observational dataset and evaluation statistics

225

230

The observational dataset considered for the model evaluation was acquired from the European Environment Agency (EEA) through the download service <a href="https://discomap.eea.europa.eu/map/fme/AirQualityExport.htm">https://discomap.eea.europa.eu/map/fme/AirQualityExport.htm</a> (last accessed, May 2023). Collected data corresponds to the E1a validated dataset where we keep only data with an hourly timestep. The E1a data are reported to EEA by member states every September, covers the year before the delivery and are considered an official delivery. Pollutants included in the evaluation are O3, NO2, PM10 and PM2.5. Then, only measurements that are considered representative of scale that the models are able to simulate (i.e., rural, suburban and urban background air pollution) are kept (not industrial or traffic proximity stations). To operate such a filter, we select background stations that are classified from 1 to 7 according to Joly and Peuch (2012) classification. In addition, observations above a certain threshold are considered outliers and removed. This threshold differs according to the pollutant and equal to 500 μg.m<sup>-3</sup> for O3, 700 μg.m<sup>-3</sup> for NO2, 1000 μg.m<sup>-3</sup> for PM10 and 700 μg.m<sup>-3</sup> for PM2.5. These pollutant-specific thresholds were defined on the basis of probability distributions of concentrations measured in Europe over 8 years, to discard potentially spurious values outside the distribution. The complete list of air quality monitoring stations used for the evaluation of the modelling results is provided in Table S2.

Modelled and observed average hourly, weekly and monthly cycles of pollutant concentrations were computed per season (i.e., January-February-March, JFM; April-May-June, AMJ; July-August-September, JAS; October-November-December, OND) to assess the benefit of the corresponding temporal profiles. For each cycle, the spatial median of the temporal correlation was estimated for the hourly/monthly/daily mean and daily maximum concentrations. The primary focus is on the correlation coefficient as temporal profiles mainly influence variability. However, since the modification of temporal profiles can also impact absolute concentration values, additional metrics—Root Mean Square Error (RMSE) and Mean Bias (MB) — were computed for daily maximum and daily mean concentrations, categorized by concentration intervals. Diagnostics are provided for all individual CAMS regional models (Table 1) and the median ensemble (ENS). To automate the evaluation process, the Python package evaltools (https://opensource.umr-cnrm.fr/projects/evaltools/wiki, last accessed: March 2025) was used. This

<sup>(2)</sup> Day types are weekday (Monday to Friday), Saturday and Sunday

package is specifically designed to evaluate predictive models of surface atmospheric composition against in-situ observations, and it is used for the evaluation of CAMS air quality models.

# 2.5 Experimental setup

To assess the impact of updating emission temporal profiles on modelled concentrations, each individual CAMS model performed two annual simulations for the meteorological year 2018. The year 2018 was chosen by convenience due to a previous modeling exercise involving several models of the CAMS ensemble (Timmermans, 2021). Furthermore, 2018 was an interesting year from a scientific point of view due to the occurrence of summer episodes of O<sub>3</sub> air pollution linked to heat waves and intense summer droughts in Europe (e.g., Pope et al., 2023).

]

245

255

260

240

Both experiments were run on a European domain (25°W-45°E, 30°N-72°N) with a 0.2°×0.2° (SILAM, MONARCH, MINNI, CHIMERE, LOTOS-EUROS) and 0.1°×0.1° (EMEP, DEHM, EURAD-IM, MATCH, GEM-AQ and MOCAGE) horizontal resolution and using global meteorological and chemical boundary and initial conditions produced with the ECMWF Integrated Forecasting System (IFS) (Flemming et al., 2015). The simulations used the same anthropogenic (CAMS-REG-AP\_v4.2 inventory for year 2017), biomass burning (GFASv1.2 for the year 2018) and other natural emissions (model-dependent). No assimilation or data fusion techniques were applied to the modelled results. In the first experiment (hereinafter referred to as expA) all models used their default set of emission temporal profiles (Table 2), while in the second experiment (hereinafter referred to as expB) they used the CAMS-REG-TEMPO dataset. For the year-dependent CAMS-REG-TEMPO profiles (Table 3), the weight factors corresponding to the year 2018 were applied.

# 250 3 Results

# 3.1 Emissions

Figure 1 to Figure 6 compare the monthly, weekly and hourly emission temporal distributions for key pollutants (i.e., NOx, NMVOC, SOx, NH<sub>3</sub>, PM<sub>10</sub>, PM<sub>2.5</sub>) across different sectors at the EU27 plus UK and Norway level using the CAMS-REG-TEMPO, TNO and GENEMIS profiles. These distributions were obtained by applying each temporal profile dataset to the CAMS-REG-AP\_v4.2 emissions. Since the total annual emissions remain the same in all three cases, the comparison focuses on differences in temporal allocation. Tables 4 to 6 summarise the correlation coefficients between monthly, weekly and hourly emissions at the EU27 plus UK and Norway estimated using CAMS-REG-TEMPO versus TNO and CAMS-REG-TEMPO versus GENEMIS per pollutant. Relative differences [%] in emission distributions— CAMS-REG-TEMPO vs. TNO and CAMS-REG-TEMPO vs. GENEMIS—are summarized per pollutant by month-of-the-year, day-of-the-week and hour-of-the-day in the Supplementary Material (Figures S1 to S3). To complement the analysis performed at the European scale, monthly, weekly and hourly correlation coefficients per individual country and pollutant are provided in Fig. S4 to S6. For hourly emission cycles (Section 3.1.3), we excluded GENEMIS from the analysis, as they report the exact same sector-dependent

hourly profiles as TNO. Instead, an additional dataset was included in the comparison: the default hourly temporal factors used in EMEP and CHIMERE, which combine GENEMIS hourly profiles (identical to TNO profiles) with the road transport profiles from Menut et al., (2012). We refer to this dataset as GENEMIS-Menutetal2012.

# 3.1.1 Monthly emission cycles





The seasonality of NO<sub>x</sub> emissions is mainly dominated by the road transport (44.3% of total emissions) and industry sectors (energy and manufacturing, 33.5% of total emissions) (Fig. 1 and Fig. S1). The monthly cycles obtained with the three temporal profile databases present correlations of 0.67 (CAMS-REG-TEMPO versus TNO) and 0.79 (CAMS-REG-TEMPO versus GENEMIS) (Table 4), with differences ranging between -10 and 10% depending on the month (Fig. S1). CAMS-REG-TEMPO presents larger emissions in February, March, July, August and November compared to the other datasets. The differences in July and August are mainly attributed to the off-road transport sector (GNFR\_I, included in the "Others" category), which CAMS-REG-TEMPO assumes to increase during summer following with the guidelines provided by EMEP/EEA (2019), whereas TNO and GENEMIS consider a flat profile due to lack of more detailed information. In February, March and November, the differences are mainly related to the meteorology-dependent profiles used in CAMS-REG-TEMPO for diesel exhaust road transport (GNFR\_F2) and residential/commercial combustion (GNFR\_C). These profiles result in a stronger contrast between cold and warm months, leading to higher emissions during colder periods compared to the profiles used in TNO and GENEMIS, which do not offer year-specific weight factors based on meteorological data.

For NMVOC (Fig. 1 and Fig. S1), the differences in monthly emissions between CAMS-REG-TEMPO and TNO are relatively small (ranging between -10% and 10%), the correlation coefficient between monthly emissions estimated by each dataset being 0.79 (Table 4). This similarity is partly due to both datasets using the same monthly profile for the solvents sector (GNFR\_E). Larger discrepancies are observed when comparing CAMS-REG-TEMPO and GENEMIS (correlation coefficient of 0.63), with the former reporting significantly lower emissions in spring —up to 20% lower in April (Fig. S1). This discrepancy is mainly driven by the different monthly profiles considered for the agricultural emissions, which fall under the GNFR\_L category. For CAMS-REG-TEMPO, the seasonality of these emissions is linked to agricultural waste burning emissions and is derived from Klimont et al. (2017), which considered the timing and location of active fires on agricultural land in the Global Fire Emissions Database (GFEDv3.1). In GENEMIS, the profile proposed for NMVOC agricultural activities is based on statistical data on sales and application of agricultural pesticides or other agrochemicals (Friedrich and Reis, 2004).

For SO<sub>x</sub>, the monthly emission cycles are largely dominated by the industry sector (81.5% of total emissions, Fig. 1). The seasonality obtained by each temporal profile database are largely correlated (0.9 and 0.89, Table 4). Compared to winter (i.e., December, January and February), the drop in industrial emissions during summer and fall is less pronounced in CAMS-REG-TEMPO than in other datasets. Consequently, emissions in July and August are up to 20% higher compared to those obtained

derived using GENEMIS profiles (Fig. S1). Conversely, emissions in January and December tend to be lower with CAMS-REG-TEMPO, showing reduction of -5% compared to TNO and -8% compared to GENEMIS.







NH<sub>3</sub> exhibits the largest differences in monthly emission distributions (Fig. 2), especially when comparing CAMS-REG-TEMPO and TNO profiles (correlation coefficient of 0.39). CAMS-REG-TEMPO reports a distinct bi-modal seasonality, with a primary peak in April (15.3% of total emissions), mainly driven by fertilizer emissions (GNFR L), and a second lowerintensity peak in July (11.5% of total emissions), mainly linked to livestock emissions (GNFR K). Concerning fertilizer emissions, the CAMS-REG-TEMPO country-dependent profiles are based on a mosaic of datasets including the regional European emission inventories reported for Denmark and Germany by Skjoth et al. (2011), for Poland by Werner et al. (2015), for the Netherlands, France and Belgium by Backes et al. (2016a) and from the global bottom-up MASAGE NH3 inventory for the rest of the countries (Paulot et al., 2014). In contrast, the TNO profile allocates the majority of NH<sub>3</sub> emissions to March (24.2% of total emissions), the peak being mainly driven by the profile proposed for fertilizer emissions. This profile is based on the work by Asman (1992), which considered information from the year 1989 in the Netherlands about the timing of manure spreading from different animal types as well as of application of synthetic fertilizers. Using CAMS-REG-TEMPO instead of TNO leads to a decrease in emissions by more than -50% during that month and an increase above 100% in summer (Fig. S1). The GENEMIS profile is more in line with that of CAMS-REG-TEMPO (correlation coefficient of 0.78), but with a flatter distribution, allocating more emissions in winter and fewer in summer (Fig. S1). The profile reported by GENEMIS is derived from measured NH<sub>4</sub><sup>+</sup> aerosol concentrations in the Netherlands in the 90s (Friedrich and Reis, 2004). It is worth mentioning that the seasonality reported by CAMS-REG-TEMPO is well aligned with European NH<sub>3</sub> emission monthly patterns derived from satellite observations, as recently reported by Ding et al., (2024).

For PM<sub>10</sub>, all three temporal profile datasets allocate more emissions in winter than in summer (31% versus 18% on average), mainly due to the seasonality of residential and commercial combustion emissions (GNFR\_C). These emissions increase during cold months as combustion activities for space heating intensify. CAMS-REG-TEMPO allocates more emissions in January and February compared to TNO (up to +20% in February), while its estimates for these months are closely aligned with GENEMIS (differences below -5%). In November and December, CAMS-REG-TEMPO reports between 15% and 20% more PM<sub>10</sub> emissions than TNO and GENEMIS, respectively. This discrepancy mainly comes from differences in the monthly allocation of agricultural waste management emissions (GNFR\_L) across datasets. In CAMS-REG-TEMPO, these emissions peak between October and December, whereas GENEMIS assigns them between September and November. TNO, on the other hand, distributes them across two peaks of similar intensity—one in spring (March-April) and another in summer (July-August). As previously mentioned, the CAMS-REG-TEMPO profiles for agricultural waste burning were derived from Klimont et al. (2017), which considered monthly emissions computed by GFEDv3.1, while in the case of TNO, the profile for this sector is derived from monthly emissions estimated by GFAS (Kaiser et al., 2012), as detailed in Kuenen et al., (2022).

For PM<sub>2.5</sub>, the monthly cycles obtained with CAMS-REG-TEMPO and GENEMIS present a U-shape pattern, whereas TNO shows a V-shape trend (Fig. 2 and Fig. S1). This discrepancy arises from differences in the monthly profiles for residential and commercial combustion emissions. The CAMS-REG-TEMPO and GENEMIS profiles are similar, as both consider the impact of meteorology (i.e., temperature-driven variations in heating demand), while for TNO the profile is based on fuel use information from small consumers (Veldt et al., 1992). Consequently, correlation between monthly emissions derived from CAMS-REG-TEMPO and GENEMIS are larger than between CAMS-REG-TEMPO and TNO (0.89 and 0.94, respectively). Maximum differences occur in February, when CAMS-REG-TEMPO reports 20% higher emissions than TNO, and in July, where CAMS-REG-TEMPO reports 20% more emissions than GENEMIS (Fig. S1). Additionally, CAMS-REG-TEMPO shows a pronounced drop in residential and commercial combustion emissions between winter and spring, leading to lower total PM2.5 emissions compared to both TNO (-20%) and GENEMIS (-25%).


 $Figure\ 1\ Monthly\ NO_x,\ NMVOC\ and\ SO_x\ emission\ temporal\ distributions\ obtained\ per\ pollutant\ and\ sector\ at\ the\ EU27\ plus\ UK\ and\ Norway\ level\ when\ using\ the\ CAMS-REG-TEMPO,\ TNO\ and\ GENEMIS\ profiles,\ respectively.$ 

Figure 2 Same as Fig. 1 for NH<sub>3</sub>, PM<sub>10</sub> and PM<sub>2.5</sub>

Table 4 Summary of correlation coefficients between monthly emissions estimated using CAMS-REG-TEMPO versus TNO and CAMS-REG-TEMPO versus GENEMIS per pollutant and at the EU27 plus UK and Norway level

| Pollutant       | r (CAMS-REG-TEMPO versus TNO) | r (CAMS-REG-TEMPO versus GENEMIS) |
|-----------------|-------------------------------|-----------------------------------|
| NO <sub>x</sub> | 0.67                          | 0.79                              |
| NMVOC           | 0.78                          | 0.63                              |
| $SO_x$          | 0.90                          | 0.89                              |
| $NH_3$          | 0.39                          | 0.78                              |
| $PM_{10}$       | 0.80                          | 0.85                              |
| $PM_{2.5}$      | 0.89                          | 0.94                              |

# 3.1.2 Weekly emission cycles





The NO<sub>x</sub> weekly cycle (Fig. 3) obtained with CAMS-REG-TEMPO presents a significantly larger drop of emissions between weekdays and weekends (-38%) compared to TNO (-22%) and GENEMIS (-15%). As a result, Saturday and Sunday emissions in CAMS-REG-TEMPO are 11% and 21% lower than those obtained using the TNO profiles (Fig. S2). The differences are slightly larger when compared to GENEMIS (-18% on Saturday and -25% on Sunday). Conversely, emissions during weekdays are between 5% and 10% higher with CAMS-REG-TEMPO than with the other datasets. These discrepancies are mainly driven by differences in the weekly profiles for road transport, which present a 44% contribution to total NOx emissions at EU27 plus UK and Norway scale, and, to a lower extent, for off-road transport (included in the "Others" category), which contribution is of 10%. As indicated in Sect. 2.3, the TNO weekly road transport profiles are based on a long time series of Dutch traffic count statistics compiled between 1985 and 1998, while the CAMS-REG-TEMPO profiles are based on TomTom congestion statistics. For off-road transport, both GENEMIS and TNO propose a default flat profile due to lack of more detailed information, while CAMS-REG-TEMPO considers the profile reported by the EMEP/EEA emission inventory guidebook (EMEP/EEA, 2019), which assume a decrease of activity from this source during weekends. The correlations between weekly emissions obtained with each dataset are very large both at the European scale (0.99 and 0.97, Table5) and across all individual countries (larger than 0.95 in all cases, Fig. S5).

For NMVOC, the weekly distributions in CAMS-REG-TEMPO and TNO are nearly identical (correlation coefficients of 1 at European scale, Table 5, and larger than 0.95 across all countries, Fig.S5), with differences ranging between -2% and 2%, depending on the day of the week (Fig. 3 and Fig. S2). Slight discrepancies are observed when comparing CAMS-REG-TEMPO and GENEMIS (correlation coefficients of 0.97, Table 5), with the former reporting 12.5% lower emissions on Saturdays and 10% higher emissions on Sundays. These differences are linked to variations in the weekly profiles for the solvent use sector. Both CAMS-REG-TEMPO and TNO use the same profile for the solvent use sector, the corresponding emissions experiencing a sharp drop between Friday and Saturday (-58% reduction), followed by stable emissions throughout the weekend. The profile is based on production and working time information from the industrial solvent use sector as reported by Lenhart and Friedrich (1995). In contrast, GENEMIS presents a gradual decline between Friday and Sunday (reduction of -77% between the two days).

The SO<sub>x</sub> weekly cycles in CAMS-REG-TEMPO and GENEMIS are almost identical (correlation coefficients of 1, Table 5), both showing a very slight drop in emissions over weekends compared to weekdays (-18%, Fig. 3). The TNO profile shows a more pronounced weekend drop, with CAMS-REG-TEMPO reporting 6% higher emissions on Saturdays and 4% higher on Sundays compared to TNO (Fig. S2). While TNO assumes a weekend effect in the residential and commercial combustion activities (GNFR\_C) due to changes in households and commercial activities as reported by Friedrich and Reis (2004), both

CAMS-REG-TEMPO and GENEMIS report a flat profile for this sector, as emissions are assumed to vary due to changes in outdoor temperature and therefore no weekend effect is considered.

Unlike the large discrepancies observed in NH<sub>3</sub> monthly cycles, the weekly cycles reported by CAMS-REG-TEMPO, TNO and GENEMIS for this species are almost identical (correlations coefficients of 0.92 and 0.93, Table 3), with all three datasets assuming a near-flat weekly distribution of emissions (Fig. 3).


For PM<sub>10</sub> and PM<sub>2.5</sub>, similar discrepancies are observed across datasets (Fig. 3). Compared to TNO, CAMS-REG-TEMPO reports slightly lower emissions on weekdays (up to -2.5%) and higher emissions on weekends (up to 7.5%) (Fig. S2). Conversely, when compared to GEMINIS, CAMS-REG-TEMPO shows higher weekday emissions (up to 4%) and lower weekend emissions (up to -7.5%). For both pollutants, these differences are mainly driven variations in the weekly profiles for the road transport (GNFR\_F) and off-road transport (included in the "Others" category) sectors across the datasets, following with what has been previously discussed for NO<sub>x</sub>. It is also important to note that for the residential and commercial combustion emissions, both CAMS-REG-TEMPO and GENEMIS do not consider a weekend effect as emissions vary according to heating degree days, while in TNO a -26% drop of emissions during weekends is assumed. Correlations between weekly PM<sub>10</sub> and PM<sub>2.5</sub> emissions are very large both at the European scale (between 0.95 and 1, Table5) and across most of the individual countries, except in those where PM emissions are primarily dominated by residential combustion emissions, where correlations are around 0.5 (e.g. Romania, Hungary, Fig. S5).

Figure 3 Weekly NO<sub>x</sub>, NMVOC and SO<sub>x</sub> emission temporal distributions obtained per pollutant and sector at the EU27 plus UK and Norway level when using the CAMS-REG-TEMPO, TNO and GENEMIS profiles, respectively.

Figure 4 Same as Fig. 3 for NH<sub>3</sub>, PM<sub>10</sub> and PM<sub>2.5</sub>

Table 5 Summary of correlation coefficients between weekly emissions estimated using CAMS-REG-TEMPO versus TNO and CAMS-REG-TEMPO versus GENEMIS per pollutant and at the EU27 plus UK and Norway level

| Pollutant       | r (CAMS-REG-TEMPO versus TNO) | r (CAMS-REG-TEMPO versus GENEMIS) |
|-----------------|-------------------------------|-----------------------------------|
| NO <sub>x</sub> | 0.99                          | 0.97                              |
| NMVOC           | 1.00                          | 0.97                              |
| $SO_x$          | 1.00                          | 1.00                              |
| $NH_3$          | 0.92                          | 0.93                              |
| $PM_{10}$       | 1.00                          | 0.97                              |
| $PM_{2.5}$      | 1.00                          | 0.95                              |

# 3.1.3 Hourly emission cycles



The NO<sub>x</sub> hourly distributions obtained with CAMS-REG-TEMPO, TNO and GENEMIS-Menutetal 2012 profiles (combination 415 of GENEMIS and the road transport profiles from Menut et al., 2012, as detailed in Section 3.1) all present a morning and afternoon peak, mainly driven by the diurnal variation of road transport emissions (Fig. 5). However, the timing and intensity of these peaks vary significantly across datasets, especially when comparing CAMS-REG-TEMPO with GENEMIS-Menutetal 2012 (correlation coefficient of 0.82 at European scale, Table 6, and close or below 0.5 for 10 individual countries, Fig. S6). Morning peak is much more pronounced in CAMS-REG-TEMPO, with total NO<sub>x</sub> emissions being approximately 420 25% higher at 07:00 and 08:00h local time (LT) compared to GENEMIS-Menutetal 2012 (Fig. S3). For the afternoon peak, significant differences exist in both intensity and timing. In CAMS-REG-TEMPO, the peak occurs between 17:00 and 18:00 LT, whereas in GENEMIS-Menutetal 2012, emissions increase more gradually and peak later, between 19:00 and 20:00 LT. Consequently, NO<sub>x</sub> emissions in CAMS-REG-TEMPO are 30-45% higher than in GENEMIS-Menutetal 2012 during 17:00h-18:00h LT. Conversely, night-time NO<sub>x</sub> emissions in CAMS-REG-TEMPO are between 30 and 50% lower than in GENEMIS-425 Menutetal 2012. The main reason behind these large discrepancies is in the design of the road transport profiles. While CAMS-REG-TEMPO were constructed considering traffic congestion statistics (see Sect. 2.3.2), in GENEMIS-Menutetal2012 profiles rely on measured NO<sub>2</sub> concentrations in urban traffic stations, which diurnal variation is controlled not only by road transport emissions but also by other physical and chemical processes not related to traffic activity, such as boundary layer dynamics and NO<sub>x</sub> titration (Li et al., 2021). The comparison between CAMS-REG-TEMPO and TNO highlights smaller 430 discrepancies in peak intensity and timing, the correlation coefficient being close to 1 at the European level (Table 6) and larger than 0.95 across individual countries (Fig. S6). While both datasets show similar peak structures, CAMS-REG-TEMPO reports slightly higher emissions, with morning peak emissions (~07:00-08:00 LT) being 5% higher than those in TNO. Nighttime NO<sub>x</sub> emissions are about 15% lower in CAMS-REG-TEMPO compared to TNO, mainly due to differences in offroad transport sector assumptions: while TNO proposes a flat profile due to lack of more detailed information, CAMS-REG-435 TEMPO concentrates most off-road emissions during daytime, following the information reported by EMEP/EEA (2019).

For NMVOC (Fig. 5), a pattern similar to that observed for NO<sub>x</sub> emissions emerges, with CAMS-REG-TEMPO allocating less emissions during night-time (between -5% and -20%) and more during daytime (between 5% and 10% (Fig. S3). However, differences are less pronounced than for NO<sub>x</sub> as all three datasets consider the same hourly profile for the dominant sector — solvent use (GNFR\_E), which translates into correlation coefficients between hourly emissions of 1 (CAMS-REG-TEMPO versus TNO) and 0.99 (CAMS-REG-TEMPO versus GENEMIS-Menutetal2012) at the European scale and also across the majority of individual countries (Fig. S6). The higher emissions in CAMS-REG-TEMPO during daytime is mainly linked to three factors; first, off-road transport emissions increase during daytime following with the information reported by EMEP/EEA (2019); second, the diurnal distribution of gasoline evaporative emissions (GNFR\_F4, included in the "road transport" category) peaks around noon due to the influence of temperature as indicated by EMEP/EEA (2019); which is not

considered in the TNO and GENEMIS hourly profiles, and third, the hourly profile for agricultural waste burning emissions peaks around noon, following with the profile proposed by Mu et al. (2011), where climatological mean hourly cycles were constructed using GOES WF\_ABBA (Geostationary Operational Environmental Satellite Wildfire Automated Biomass Burning Algorithm) active fire satellite observations.





For SO<sub>x</sub> (Fig. 5), differences in hourly emission cycles are rather small (correlation coefficients of 1, Table 6). CAMS-REG-TEMPO shows a flatter distribution of industrial emissions, resulting in a smaller contrast between night-time (23:00 to 06:00) and daytime (07:00 to 22:00) total SO<sub>x</sub> emissions (-28% reduction between night- and daytime) when compared to TNO and GENEMIS-Menutetal2012 (-34% reduction). As presented in Fig. S3, CAMS-REG-TEMPO reports lower SO<sub>x</sub> emissions between 07:00 and 17:00h LT (approx. -5%) and higher emissions between 18:00 till 06:00h LT (between 2% and 8% compared to both the TNO and GENEMIS-Menutetal2012 profiles (Fig. S3). These discrepancies are due to the different profiles considered for the public power sector (GNFR\_A, included in the "Industry" category in Fig. 5). While TNO and GENEMIS rely on information from the 90s on fuel use and load curves from power plants (Friedrich and Reis, 2004), CAMS-REG-TEMPO country-dependent profiles are based on electricity production statistics compiled from the European Network of Transmission System Operators for Electricity (ENTSO-E; Hirth et al., 2018) for the years 2015-2017.

Similar to SO<sub>x</sub>, differences in NH<sub>3</sub> diurnal cycles are minimal (<5%, Fig. 6 and Fig. S3), as all three datasets consider the same hourly profiles for agriculture and livestock emissions, the two dominant sources of NH<sub>3</sub>. The profile is derived from the work by Asman (1992), which determined the diurnal evolution of NH<sub>3</sub> emissions as a function of the variation in the soil temperature, which has a large influence on the NH<sub>3</sub> concentration at the soil surface, and the variation in the atmospheric turbulence, which determines the maximum rate at which the NH<sub>3</sub> at the soil surface can be transported to the air. Two climatological data sets obtained from measurement stations in the Netherlands (De Bilt) and Denmark (Kastrup) where used to compute the diurnal variation of NH<sub>3</sub> emissions considering the aforementioned influences. The hourly profile considered in the present datasets is the results of averaging the annually averaged relative diurnal variations obtained in the two locations. As a result of applying the same profile for agricultural and livestock sources, correlations between hourly emissions are 1 both at the European scale (Table 6) and across individual countries (Fig. S6)

Finally, large discrepancies are observed in the diurnal distributions of PM<sub>10</sub> and PM<sub>2.5</sub> (Fig. 6). CAMS-REG-TEMPO reports much higher emissions during the evening hours (17:00- 22:00h LT). This discrepancy is mainly driven by differences in the hourly distribution of residential and commercial combustion emissions (GNFR\_C). In CAMS-REG-TEMPO, these emissions, largely linked to residential wood combustion in fireplaces, boilers and other types of appliances, are assumed to peak in the evening based on the information derived from citizen interviews in Norway and Finland (Finstad et al., 2004 and Gröndahl et al., 2010) as well as from measurements of the wood-burning fraction of black carbon in Athens (Athanasopoulou et al., 2017). In contrast, TNO and GENEMIS-Menutetal2012 distribute emissions more evenly, with two peaks: one in the

morning and another in the afternoon, as the hourly profile for this sector is only based on household gaseous fuel consumption statistics (Friedrich and Reiss, 2004). Consequently, PM emissions in CAMS-REG-TEMPO are over 50% higher than those in TNO and GENEMIS-Menutetal2012 between 17:00 and 19:00h LT, while morning peak emissions are approximately 40% lower. Due to the differences in the hourly profile considered for the residential and commercial sector, correlations of total hourly emissions are lower than the ones observed for the other primary pollutants (0.57, Table 6).

Figure 5 Diurnal  $NO_x$ , NMVOC and  $SO_x$  emission temporal distributions obtained per pollutant and sector at the EU27 plus UK and Norway level when using the CAMS-REG-TEMPO, TNO and GENEMIS-Menutetal2012 profiles, respectively.

Figure 6 Same as Fig. 5 for NH<sub>3</sub>, PM<sub>10</sub> and PM<sub>2.5</sub>

Table 6 Summary of correlation coefficients between hourly emissions estimated using CAMS-REG-TEMPO versus TNO and CAMS-REG-TEMPO versus GENEMIS-Menutetal2012 per pollutant and at the EU27 plus UK and Norway level

| Pollutant       | r (CAMS-REG-TEMPO versus TNO) | r (CAMS-REG-TEMPO versus GENEMIS-Menutetal2012) |
|-----------------|-------------------------------|-------------------------------------------------|
| NO <sub>x</sub> | 0.99                          | 0.82                                            |
| NMVOC           | 1.00                          | 0.99                                            |
| $SO_x$          | 1.00                          | 1.00                                            |
| $NH_3$          | 1.00                          | 1.00                                            |
| $PM_{10}$       | 0.68                          | 0.67                                            |
| $PM_{2.5}$      | 0.57                          | 0.57                                            |

# 3.2 Correlation of modelled diurnal, weekly and monthly cycle concentrations with surface observations







Figure 7 summarises the differences in temporal correlation values obtained by the ENS in expB (CAMS-REG-TEMPO profiles) and expA (default profiles). Results are provided per species, cycle type (monthly, weekly and diurnal) and season. Positive values indicate improvements in correlation when using CAMS-REG-TEMPO, while negative values (red boxes) indicate degradations. Absolute changes in correlation between -0.01 and 0.01 are considered insignificant (grey boxes). The values in brackets indicate the maximum and minimum correlation differences obtained across the individual CAMS regional models. The results for each individual model are provided in the Supplementary Material (Fig. S7). Please note that due to technical issues during the simulations, the modelled concentrations of DEHM (MATCH) NO<sub>2</sub> and O<sub>3</sub> (PM<sub>10</sub> and PM<sub>2.5</sub>) were excluded from the comparative analysis and are therefore not available in the supplementary material.

O<sub>3</sub> is the pollutant with the lowest sensitivity to changes in temporal profiles. For both monthly and diurnal cycles (all seasons), correlation values remain almost unchanged when moving from the default (TNO, GENEMIS) to CAMS-REG-TEMPO profiles. Note that for these two cycles the correlation values of the ENS are also the largest among the four species analysed (between 0.90 and 0.95, see Sect. 3.2.2 for more details) and therefore the room for improvement is very limited. At the weekly level, the impact varies by season. During JFM and OND, slight correlation improvements are observed (+0.03 and +0.02), whereas during AMJ and JAS, degradations of -0.1 and -0.03, respectively, are reported. These degradations clearly contrast with the improvements in NO<sub>2</sub> weekly cycles observed during the same seasons (+0.13 for AMJ and +0.08 for JAS).

 $NO_2$  exhibits the largest variation in temporal correlation due to CAMS-REG-TEMPO, with only minor degradations occurring in the diurnal cycle during AMJ (-0.03). The improvements in  $NO_2$  weekly correlations are consistent across all models except for MATCH, which largely increases the correlations during AMJ (0.19) and JAS (0.49) but also shows slight degradations during JFM (-0.09) and OND (-0.08). Overall, differences between expB and expA reach up to +1.0 (see Sect. 3.2.1 for more details).

For PM<sub>10</sub>, the major improvement occurs in the OND diurnal cycle (+0.13), the JFM diurnal and JAS weekly cycles also showing a slight improvement (+0.02 in both cases), while a minor degradation is reported for the AMJ weekly cycle (-0.03). PM<sub>10</sub> is also the only pollutant to show a slight improvement in the monthly cycle correlation (+0.02), while other pollutants showing no changes. As shown in Sect. 3.2.1 to 3.2.4, the monthly correlations reported by the ENS in expA for O<sub>3</sub>, NO<sub>2</sub> and PM<sub>2.5</sub> are already very high (0.95, 0.83 and 0.82, respectively), while PM<sub>10</sub> presents the lowest correlation (0.68), giving more room for improvement.

Similarly, PM<sub>2.5</sub> shows a major correlation improvement in the OND diurnal cycle (+0.15), mirroring PM<sub>10</sub>. The OND weekly cycle shows a slight improvement (+0.02), while for other seasons correlation values remain either unchanged (monthly and all weekly cycles except AMJ) or show slightly degradations (JFM and AMJ diurnal cycles: -0.02; AMJ weekly cycle: -0.04).







Overall, the sensitivity to changes in the emission temporal profiles is larger for  $NO_2$  and  $PM_{10}$ , which are dominated by primary sources, and lower for  $PM_{2.5}$  and  $O_3$ , which are primarily driven by secondary formation and, in the case of  $O_3$ , by remote influences due to its higher lifetime.

| n allutant        | diurnal        |                |                | weekly         |                |                 |                | monthly.       |                |
|-------------------|----------------|----------------|----------------|----------------|----------------|-----------------|----------------|----------------|----------------|
| pollutant         | JFM            | AMJ            | JAS            | OND            | JFM            | AMJ             | JAS            | OND            | monthly        |
| $O_3$             | 0.00           | 0.00           | 0.00           | 0.00           | 0.03           | -0.10           | -0.03          | 0.02           | 0.00           |
| 03                | (-0.04 / 0.02) | (-0.02 / 0.02) | (-0.01 / 0.02) | (-0.04 / 0.02) | (0.00 / 0.03)  | (-0.13 / -0.02) | (-0.04 / 0.01) | (-0.03 / 0.09) | (-0.03 / 0.00) |
| NO <sub>2</sub>   | 0.05           | -0.03          | 0.00           | 0.07           | 0.16           | 0.13            | 0.08           | 0.06           | 0.00           |
| 1102              | (-0.10 / 0.19) | (-0.18 / 0.1)  | (-0.15/0.07)   | (-0.11 / 0.24) | (-0.09 / 0.44) | (0.01 / 1.0)    | (0.01 / 0.76)  | (-0.08 / 0.17) | (-0.01 / 0.1)  |
| PM <sub>10</sub>  | 0.02           | -0.01          | 0.01           | 0.13           | 0.01           | -0.03           | 0.02           | 0.00           | 0.02           |
| r 1 <b>v1</b> 10  | (-0.01 / 0.32) | (-0.2 / 0.02)  | (-0.07 / 0.06) | (0.04 / 0.29)  | (-0.03 / 0.07) | (-0.07 / 0.03)  | (-0.01 / 0.17) | (-0.02 / 0.03) | (-0.08 / 0.07) |
| DM                | -0.02          | -0.02          | -0.01          | 0.15           | 0.00           | -0.04           | 0.00           | 0.02           | 0.00           |
| PM <sub>2.5</sub> | (-0.08 / 0.49) | (-0.36 / 0.02) | (-0.13 / 0.11) | (0.03 / 0.51)  | (-0.08 / 0.06) | (-0.13 / 0.02)  | (-0.01 / 0.1)  | (-0.03 / 0.04) | (-0.09 / 0.1)  |

Figure 7 Summary of the ENS correlation differences (expB – expA) per species (O<sub>3</sub>, NO<sub>2</sub>, PM<sub>10</sub>, PM<sub>2.5</sub>), season (JFM, AMJ, JAS, OND) and cycle (diurnal, weekly, monthly). Values between brackets indicate the minimum and maximum correlation differences among the individual CAMS regional models. Boxes highlighted in green/salmon/grey indicate an improvement/degradation/no significant changes (between -0.01 and 0.01) in the correlation when using CAMS-REG-TEMPO.

Figure 8 illustrates the ENS correlation differences (expB – expA) at the station level, categorized by species (O<sub>3</sub>, NO<sub>2</sub>, PM<sub>10</sub>, PM<sub>2.5</sub>) and selected seasons. Each species is analysed during the season when its concentrations are at their maximum levels. For NO<sub>2</sub> a general improvement in correlation during JFM is observed across the domain. In contrasts, O<sub>3</sub> during JAS shows more heterogeneous results, with improvements in central Europe (e.g., Germany) and degradations in western (e.g. Spain, France) and eastern (e.g. Poland) countries. One aspect that is interesting to highlight about the slight deterioration of the scores in Western Europe is that it mainly affects rural areas (as opposed to urban areas). This is clearly visible for France and Spain, where we can see that in stations located in the respective capitals (Paris and Madrid) and other urban areas (Marseille, Barcelona) correlations are increasing, while in rural regions scores are being deteriorated. These results highlight the added value of the new CAMS-REG-TEMPO profiles for areas with high NO<sub>x</sub> emissions, particularly the profiles proposed for the road transport sector, which is the main dominant source of NO<sub>x</sub> emissions in urban areas. Since the deterioration is mainly occurring in rural areas, one hypothesis to explain these results could be the potential influence of the online biogenic NMVOC and soil NO<sub>x</sub> emission parametrisations considered in each CAMS model, as described in Colette et al. (2024). Downwind urban areas other processes like meteorology and photochemistry may dominate the signal. For PM<sub>10</sub>, JFM correlations improve at stations in Germany, Poland, Portugal and parts of Spain, whereas degradations are observed in France and the Czech Republic. Conversely, for PM<sub>2.5</sub> during OND, France reports more stations with improved correlations, while Germany exhibits a decrease in most sites.

Figure 8 Summary of the ENS correlation differences (expB - expA) at the station level per species (O<sub>3</sub>, NO<sub>2</sub>, PM<sub>10</sub>, PM<sub>2.5</sub>) and selected seasons. Green values indicate an improvement in the correlations when using CAMS-REG-TEMPO, while red values indicate a degradation.

# 3.2.1 $NO_2$

There is no significant variation in the ENS correlation coefficient for the NO<sub>2</sub> monthly cycle when using CAMS-REG-TEMPO (0.83 versus 0.84). However, its implementation induces a consistent positive response across most individual models (7 models), with correlation increases ranging from +0.09 (CHIMERE) to +0.004 (EMEP) (Fig. 9). Notably, the ENS captures better the observed NO<sub>2</sub> peak in February (Fig. 9). This improvement is likely driven by the meteorology-dependent temporal profiles applied to the residential and commercial combustion and diesel road transport sectors in CAMS-REG-TEMPO. These profiles lead to an increase of the total NO<sub>x</sub> emissions during February when compared to TNO (10%) and GENEMIS (2%), as shown in Fig. S1, reflecting the Hartmut cold spell, a winter storm that brought a cold wave and negative temperature anomalies to large areas of Europe during that month (C3S, 2018).

The largest improvement in NO<sub>2</sub> correlation is observed in the weekly cycle across all individual models (Fig. 10 and Fig. S8). For the ENS, correlation increases from 0.66 to 0.82 (+0.16) in JFM, 0.66 to 0.80 (+0.14) in AMJ, 0.78 to 0.86 (+0.08) in JAS and 0.82 to 0.88 (+0.06) in OND, exceeding 0.8 for all four seasons. This improvement is consistent across all individual models except for MATCH in JFM and OND, during which slight degradations are reported (-0.09 and -0.08, respectively). The effect is especially pronounced in models that previously used GENEMIS profiles in the expA (i.e., EMEP and CHIMERE), showing substantial correlation increases —up to +1.00 in AMJ (from -0.21 to 0.79) and +0.76 in JAS (from 0.10 to 0.86). The ENS improvement is mainly due to a better reproduction of the observed weekday-to-weekend drop in NO<sub>2</sub> concentrations when using CAMS-REG-TEMPO. As discussed in Sect. 3.1.2, the TomTom congestion-derived profiles used in CAMS-REG-TEMPO for the road transport sector result in larger weekday-to-weekend differences in NO<sub>x</sub> emissions (-38%), particularly compared to the GENEMIS profiles (-15%).

For the NO<sub>2</sub> diurnal cycle, results vary considerably depending on the model and season (Fig. 11 and Fig. S9). In expA, correlation values for the ENS range between 0.64 to 0.75. A slight positive impact is observed for the ENS and 7 of CAMS individual models during JFM and OND (+0.05 and +0.07 for the ENS, respectively), when NO<sub>2</sub> levels are at their maximum, while no changes are observed during JAS. Conversely, a slight degradation occurs during AMJ (-0.03), mainly due to changes in the intensity of the morning (6-8 a.m.) and evening (6-8 p.m.) peaks in the diurnal cycle. It is important to note that the temporal emission profiles in expA are not uniform across all models (Table 2), which partly explains the heterogeneous results. However, even among models using the same profiles in expA, contrasting results emerge when switching to CAMS-REG-TEMPO profiles. For instance, while MONARCH and MINNI show consistent improvements across all four seasons (correlation values increasing from +0.02 to +0.20), LOTOS-EUROS correlations are consistently degraded (decreases from -0.02 to -0.23), despite all three models using TNO profiles in expA. Similarly, while CHIMERE shows significant correlation improvements in all seasons ranging from +0.06 in AMJ and +0.24 in OND, EMEP reports only slight improvements in JFM (+0.04) and OND (+0.03), even though both models use the GENEMIS profiles in expA. This heterogenous impact illustrates

the complex interactions between emission temporal distributions and other model-related processes, such as the planetary boundary layer depth cycle.

Figure 9 Comparison between the observed and modelled NO<sub>2</sub> monthly cycle for the ENS (left) and spatial median of the temporal correlations obtained for the ENS and each individual CAMS model in expA (red) and expB (blue).

Figure 10 Comparison between the observed and modelled  $NO_2$  weekly cycle for the ENS for JFM and JAS (left) and spatial median of the temporal correlations obtained for the ENS and each individual CAMS model per season in expA (red) and expB (blue). The JFM and JAS periods were selected because they represent winter-like and summer-like conditions as well as the highest and lowest values of the year.


Figure 11 Comparison between the observed and modelled NO<sub>2</sub> diurnal cycle for the ENS (UTC time) for JAS and OND (left) and spatial median of the temporal correlations obtained for the ENS and each individual CAMS model per season in expA (red) and expB (blue). JAS and OND periods were selected because they represent periods were ENS show an improvement and deterioration of the correlation when using CAMS-REG-TEMPO, respectively.

# 3.2.2 $O_3$





For the ENS and most individual models (10 out of 11), the correlation coefficient of the O<sub>3</sub> monthly cycle is already high (above 0.9) and shows little sensitivity to the implementation of the CAMS-REG-TEMPO profiles (less than 0.005 changes in the correlation between expA and expB, Fig. 12). In contrast, the weekly cycle is impacted (Fig. 13 and Fig. S8). On average, slight correlation improvements are observed for JFM (+0.03 for the ENS) and OND (+0.02), while decreases occur in AMJ (-0.10) and JAS (-0.03). This behaviour is generally consistent across all individual models except for EMEP, which presents an improvement of the weekly correlation for all four seasons (Fig. S7). During JFM, the use of CAMS-REG-TEMPO enhances the models' ability to capture the O<sub>3</sub> weekend effect —increase of O<sub>3</sub> concentrations during weekends due to reduced NO<sub>x</sub> emissions, which limits O<sub>3</sub> titration. However, in JAS, this effect is slightly degraded with CAMS-REG-TEMPO, despite NO<sub>2</sub> correlation improvements during the same season. This illustrates the complexity of the O<sub>3</sub> cycle, which exhibits nonlinear relationships with its main precursors, NO<sub>x</sub> and VOCs. Similar to the monthly cycle, the diurnal cycle correlation coefficient remains largely unchanged across all seasons (Fig. 14 and Fig. S9). The ENS and all the individual models consistently show strong performance in reproducing the observed O<sub>3</sub> diurnal cycle, especially during AMJ and JAS (ENS correlation: 0.95), when concentrations are at their maximum. The low sensitivity of O<sub>3</sub> modelled cycle concentrations to changes in the emission temporal profiles can also be partially explained by the importance of O<sub>3</sub> hemispheric contributions to European background levels (Garatachea et al., 2024). We attribute the positive bias of O<sub>3</sub> nighttime levels reported in Fig. 14 to the negative bias of the modelled NO<sub>x</sub> levels (Fig. 11), which lead to an underestimation of O<sub>3</sub> loss via NO titration. The O<sub>3</sub> nighttime overestimation is a common feature of air quality models and has been extensively discussed in previous works (e.g., Bessagnet et al., 2016; Pay et al., 2019).

Figure 12 Comparison between the observed and modelled O<sub>3</sub> monthly cycle for the ENS (left) and spatial median of the temporal correlations obtained for the ENS and each individual CAMS model in expA (red) and expB (blue).

Figure 13 Comparison between the observed and modelled O<sub>3</sub> weekly cycle for the ENS for JFM and JAS (left) and spatial median of the temporal correlations obtained for the ENS and each individual CAMS model per season in expA (red) and expB (blue). The JFM and JAS periods were selected because they represent winter-like and summer-like conditions as well as the highest and lowest values of the year.

Figure 14 Comparison between the observed and modelled O<sub>3</sub> diurnal cycle for the ENS (UTC time) for JFM and JAS (left) and spatial median of the temporal correlations obtained for the ENS and each individual CAMS model per season in expA (red) and expB (blue). The JFM and JAS periods were selected because they represent winter-like and summer-like conditions as well as the highest and lowest values of the year.

# 3.2.3 PM<sub>10</sub>

The correlation coefficient of the PM<sub>10</sub> monthly cycle shows a slight improvement in the ENS and most individual models (8 out of 11), with an increase up to 0.09 in MATCH (Fig. 15). The unrealistic peak modelled in April by expA, which is not observed in measurements, is significantly smoothed when using CAMS-REG-TEMPO profiles. This improvement is linked to a reduction of more than 20% in primary PM<sub>10</sub> emissions in April under CAMS-REG-TEMPO, compared to the default profiles. A slight degradation is observed in models using the GENEMIS profiles in expA, for which correlation decrease by -0.07 (CHIMERE) and -0.03 (EMEP). The lower correlation in these two models is related to a less accurate reproduction of the observed PM<sub>10</sub> level increases between January and February (CHIMERE) and September and October (EMEP) when moving from GENEMIS (expA) to CAMS-REG-TEMPO profiles (expB). For the first case, the degradation could be linked to the fact that NH<sub>3</sub> emissions, which largely contribute to the formation of secondary inorganic aerosols during cold months (e.g. Backes et al., 2016b; Clappier et al., 2021), remain constant between January and February when using CAMS-REG-

TEMPO (+0.5% increase), while a large increase is observed when considering the GENEMIS profiles (+14%), as reported in Fig. 2. For the second case, the reduction in accuracy could be related to the lower increase in primary PM10 emissions between September and October reported by CAMS-REG-TEMPO (32%) when compared to GENEMIS (38%), combined with the -13% decrease (9% increase) of total NH<sub>3</sub> emissions reported by CAMS-REG-TEMPO (GENEMIS) for the same period.

For weekly profiles, a consistent slight improvement is observed for ENS (+0.01) and across 7 individual CAMS models during JFM and OND (Fig. 16), when PM<sub>10</sub> concentrations are at their maximum (Fig. 15). The largest improvements are reported during JFM by EMEP (+0.07) and MINNI (+0.05). Additionally, the bias between models and observations is slightly reduced in OND (-8.3%), as CAMS-REG-TEMPO allocates approximately 20% more PM<sub>10</sub> emissions in November and December compared to the default TNO and GENEMIS profiles (Fig. S1).


Similar to NO<sub>2</sub>, the impact of CAMS-REG-TEMPO on the PM<sub>10</sub> diurnal cycle is heterogeneous across seasons (Fig. 17 and Fig. S9). A significant improvement is observed during OND in 10 individual models, with the correlation coefficient increasing by over 50% for GEM-AQ and DEHM, and by more than 25% for LOTOS-EUROS, CHIMERE and MINNI. The improvement is less pronounced during JFM (8 individual models), with correlation increases of up to 10%. During JFM and OND, CAMS-REG-TEMPO better reproduces the observed evening peak, which is typically higher than the morning peak, especially in OND. In contrast, TNO and GENEMIS profiles tend of overestimate the morning peak relative to the evening peak. The enhanced performance of CAMS-REG-TEMPO can be mainly attributed to its diurnal profiles for residential and commercial combustion emissions, which concentrates 63% of the emissions from this source in the evening (between 17:00 and 23:00h), whereas the profiles proposed by TNO and GENEMIS for this sector distribute only 32% of the emissions to this time of the day (Fig. 6). However, the use of CAMS-REG-TEMPO diurnal profiles also increases the negative bias in the modelled morning PM<sub>10</sub> peak. We partly attribute this bias to the omission of road transport resuspension emissions in the CAMS-REG-ANT inventory, as these are currently excluded in official reporting despite being reported as a significant contributor to the PM<sub>10</sub> primary emissions in Europe (e.g., Denier van der Gon et al., 2018).

A shift of approximately two hours between the modelled and measured PM<sub>10</sub> morning peak is observed both in the expA and expB ENS results. This PM peak shift problem is frequent and known for several years. As indicated by Schaap et al. (2011), this issue could be related to limitations in the reproduction of the diurnal cycles of inorganic aerosols (e.g., nitrate, sulphate, ammonium, nitric acid and ammonia). Another aspect that could be driven the shift between PM modelling results and observations are transport and/or chemical reaction pathways relevant to the formation of secondary organic aerosols that are not adequately included in chemical transport models' input or formulation, as reported by Mircea et al., (2019). Other aspects that could explain the limitations of the modelling results could be the representation of dynamic processes and the development of the boundary layer, which can be difficult to simulate in regions with complex topography with chemical

transport models running at  $\sim$ 10km resolution. Further investigations should be performed to understand the causes behind this discrepancy.

Figure 15 Comparison between the observed and modelled  $PM_{10}$  monthly cycle for the ENS (left) and spatial median of the temporal correlations obtained for the ENS and each individual CAMS model in expA (red) and expB (blue).

expA expB

- expA - expB - obs

Figure 16 Comparison between the observed and modelled PM<sub>10</sub> weekly cycle for the ENS for JFM and OND (left) and spatial median of the temporal correlations obtained for the ENS and each individual CAMS model per season in expA (red) and expB (blue). The JFM and OND periods were selected because they represent the highest values of the year.

Figure 17 Comparison between the observed and modelled  $PM_{10}$  diurnal cycle (UTC time) for the ENS for JFM and OND (left) and spatial median of the temporal correlations obtained for the ENS and each individual CAMS model per season in expA (red) and expB (blue). The JFM and OND periods were selected because they represent the highest values of the year.

# 3.2.4 PM<sub>2.5</sub>



For PM<sub>2.5</sub> there is no significant variation in the correlation coefficient for the monthly cycle in the ENS ( $\pm$ 0.01) (Fig. 18). The CAMS models using the TNO profiles by default tend to present significant improvements (up to  $\pm$ 0.10 and  $\pm$ 0.08 EURAD-IM and for MOCAGE, respectively) while a degradation is observed in those models using GENEMIS by default, with correlation decreases up to  $\pm$ 0.10 in the case of CHIMERE and  $\pm$ 0.04 in the case of EMEP. It is important to note that the EEA observational coverage for PM<sub>2.5</sub> is less comprehensive in some countries (e.g., Spain, Italy) compared to other pollutants analysed (Fig. 8), which may influence these results.

For the weekly cycle (Fig. 19 and Fig. S8), the ENS shows an average correlation decrease of -0.04 in AMJ with expB and a slight increase of +0.02 in OND. During the other two seasons (JFM and JAS), the weekly cycle correlation remains unchanged for the ENS, reflecting a balance between improvements and degradations across individual models. A total of 6 and 7 individual models reports improvements during JFM and JAS, respectively, the others reporting degradations of similar magnitude (e.g., +0.06 for DEHM vs. -0.08 for CHIMERE in JFM).

At the hourly scale (Fig. 20 and Fig. S9), results closely resemble those observed for PM<sub>10</sub>. While there is a slight correlation decrease in AMJ (-0.01 on average for the ENS), a considerable increase is observed in OND (+0.15 on average for the ENS). As mentioned in Sect. 3.2.3, this improvement is mainly driven by the diurnal profile for residential combustion emissions in CAMS-REG-TEMPO. The two hours shift between modelled and measured morning peaks is also noticeable here, as reported for PM<sub>10</sub>.

Figure 18 Comparison between the observed and modelled PM<sub>2.5</sub> monthly cycle for the ENS (left) and spatial median of the temporal correlations obtained for the ENS and each individual CAMS model in expA (red) and expB (blue).

Figure 19 Comparison between the observed and modelled PM<sub>2.5</sub> weekly cycle for the ENS for JFM and JAS (left) and spatial median of the temporal correlations obtained for the ENS and each individual CAMS model per season in expA (red) and expB (blue). The JFM and JAS periods were selected because they represent winter-like and summer-like conditions as well as the highest and lowest values of the year.

Figure 20 Comparison between the observed and modelled PM<sub>2.5</sub> diurnal cycle for the ENS (UTC time) for JAS and OND (left) and spatial median of the temporal correlations obtained for the ENS and each individual CAMS model per season in expA (red) and expB (blue). The JFM and JAS periods were selected because they represent the highest values of the year as well as periods were the ENS shows a deterioration and improvement of the correlation when using CAMS-REG-TEMPO, respectively.

# 735 3.3 Average deviation from observations



Figure 21 shows the annual spatial median of bias and RMSE computed by concentration intervals for the ENS across species for expA and expB. Overall, the statistics hardly vary between experiments, although slight decreases in both bias and RMSE are observed at higher concentration ranges when comparing expB to expA for  $O_3$  (bias and RMSE reductions of -2.4% and -1.4% for concentrations ranging from 110 to 130  $\mu$ g/m³ and of -1.1% and -0.8% for concentrations ranging from 130 to 150  $\mu$ g/m³) and PM<sub>10</sub> (bias and RMSE reductions of -2.1% and -1.3% for concentrations equal or larger than 60  $\mu$ g/m³). Concerning NO<sub>2</sub>, the larger improvements are observed for concentrations raging between 40 and 60  $\mu$ g/m³ (bias and RMSE reductions of -3.1% and -2.4%), while in the case of PM<sub>2.5</sub> the reduction of the bias and RMSE is mainly occurring at low concentration ranges (bias and RMSE reductions of -11.9% and -0.5% for concentrations ranging from 5 to 10  $\mu$ g/m³).

Figure 21 Spatial median of bias and RMSE computed by concentration intervals for the ENS per species (NO<sub>2</sub>, O<sub>3</sub>, PM<sub>10</sub> and PM<sub>2.5</sub>) in expA (red) and expB (blue).

Unlike annual averages, concentrations can vary significantly between seasons. To complement this analysis, Fig. 22 shows the spatial median of the observed and modelled (ENS, expA and expB) daily maximum concentration of O<sub>3</sub> and daily mean concentration of NO<sub>2</sub> and PM<sub>2.5</sub> for selected seasons. The selected seasons represent winter-like and summer-like conditions as well as the highest and lowest concentration values of the year. Results for the remaining seasons and for PM<sub>10</sub>, which conclusions are almost identical to the ones obtained for PM<sub>2.5</sub>, are reported in the Supplementary material (Figure S10).

For O<sub>3</sub>, which exhibits a high seasonal variation, significant differences emerge between expA and expB during AMJ and JAS when both modelled and observed concentrations are at their maximum (Fig. 12). In AMJ, the median of the daily maximum concentration is lower with expB (94.9 μg/m³) compared to expA (96.2 μg/m³), which translates into a 28% increase of the bias between the ENS and observations. This bias increase is driven by the lower NO<sub>x</sub> emissions available to enhance O<sub>3</sub> formation during April and May when using CAMS-REG-TEMPO instead of TNO or GENEMIS (approximately -10%, as indicated in Fig. S1). Conversely, during JAS, expB reports higher concentrations during July and the first half of August (104.4 and 106.9 and μg/m³ for expA and expB, respectively), reducing the bias when compared to observations, especially during the large-scale O<sub>3</sub> pollution episodes occurred between the 23<sup>rd</sup> and 27<sup>th</sup> of July (bias reduction of -29.2%) and 2<sup>nd</sup> and 7<sup>th</sup> of August (bias reduction of -23.7%). We attribute this reduction in the biases to the larger amount of NO<sub>x</sub> emissions allocated to July and August when using CAMS-REG-TEMPO with respect to TNO and GENEMIS profiles (up to +8% according to Fig. S1).

Regarding NO<sub>2</sub>, an increase of 6.2% in averaged modelled levels is observed when comparing expA with expB results for JFM, while during OND the two experiments report in average the same concentrations (7.1 μg/m<sup>3</sup>). Similarly, when looking at the day with the largest observed peak per season (8<sup>th</sup> of February for JFM and 17<sup>th</sup> of December for OND), expB only allows reducing the biases of the ENS for the JFM day (-16.4%), while for the OND day it remains almost unchanged (-2.1%).

770

780

For PM<sub>2.5</sub>, modelled daily mean concentrations in expB (9.3 μg/m³) are in average slightly higher than in expA (9.0 μg/m³) during JFM, except for the pollution episode occurred between 20<sup>th</sup> and 22<sup>nd</sup> February, when the bias of the ENS is increased by 28% when moving from expA to expB. While primary PM<sub>2.5</sub> emissions in February are 25% higher when considering CAMS-REG-TEMPO instead of TNO, emissions from NH<sub>3</sub> are more than 50% lower (Fig. S1), which may reduce the formation of fine secondary inorganic aerosols. Concentrations from expB are on average 7.3% higher during JAS when compared to expA. This behaviour can be linked to the increase in primary PM<sub>2.5</sub> emissions in July under CAMS-REG-TEMPO compared to TNO or GENEMIS profiles (approx. 20%), as reported in Fig. 2 and Fig. S1. Additionally, the rise in key precursors of secondary fine aerosols, such as NH<sub>3</sub> (up to 100% and 50% increases when compared to TNO and GENEMIS, respectively, Fig. 2 and S1), may also contribute to these differences.

Figure 22 Spatial median of the observed (black) and modelled (red, expA; blue expB) daily maximum concentration of O<sub>3</sub> by the ENS during AMJ and JAS and daily mean concentration of NO<sub>2</sub> during JFM and OND and PM<sub>2.5</sub> during JFM and JAS for 2018.

### 4 Conclusions




- This study evaluates the impact of implementing updated anthropogenic emission temporal profiles on the performance scores of the CAMS European multi-model ensemble air quality modelling system. The CAMS-REG-TEMPO emission temporal profiles dataset was compared against the default temporal distributions considered in the 11 regional models that conform the CAMS ensemble, namely the TNO and GENEMIS profiles. The sensitivity of these models plus the ensemble (ENS, median of the 11 models) was assessed by comparing the simulation results with NO<sub>2</sub>, O<sub>3</sub>, PM<sub>2.5</sub> and PM<sub>10</sub> observations from the EEA European air quality monitoring network. Model-observation comparisons were conducted for average hourly, weekly and monthly pollutant concentrations, analysed per season (JFM, AMJ, JAS, OND) to quantify the impact of CAMS-REG-TEMPO. The findings show that the effects of integrating CAMS-REG-TEMPO profiles vary depending on the pollutant and time cycle considered:
- NO<sub>2</sub> presents the greatest improvement in temporal correlation with CAMS-REG-TEMPO. The weekly cycle correlations present increases of up to +1.04 for one model and +0.06 (OND) to +0.16 (JFM) in the ENS. This improvement is mainly linked to the CAMS-REG-TEMPO weekly profiles for road transport. At the monthly scale, a better representation of the February NO<sub>2</sub> peak is observed due to the use of meteorological-dependent profiles. For the diurnal cycle, results vary considerably by model and the season. A positive impact is observed for the ENS (up to +0.07) and most models (up to +0.24) during JFM and OND, when NO<sub>2</sub> peaks. However, AMJ shows slight degradations, with correlation decreases of up to -0.18 in one model and -0.03 in the ENS.
  - O<sub>3</sub> is least affected by changes in emission temporal profiles. Monthly and diurnal cycles remain almost unchanged across seasons, as O<sub>3</sub> correlation in the ENS are already high (between 0.9 and 0.95), leaving little the room for improvement. At weekly level, small correlation improvements are observed in JFM and OND (+0.03 for the ENS), while degradations occur in AMJ (-0.10) and JAS (-0.03). This contrast with NO<sub>2</sub> highlights the complexity of O<sub>3</sub> formation, its non-linear relationship with NO<sub>3</sub> and VOCs and the importance of O<sub>3</sub> long range transport.
  - PM<sub>10</sub> is the only pollutant showing a notable improvement in the monthly cycle correlation (+0.03 in the ENS). Weekly cycle correlation improves slightly across the ENS and individual models in JFM and OND (up to +0.07), when PM<sub>10</sub> concentrations peak. Diurnal cycle results are more heterogeneous, with significant improvements in OND (+0.13 for the ENS) and moderate improvement in JFM (+0.02 in the ENS). These improvements are mainly linked to the revised diurnal profile residential and commercial combustion emissions, which better capture the observed evening peak, typically larger than the morning peak.
  - PM<sub>2.5</sub> results are more variable, depending on the model's default profiles. Models using the TNO profiles by default show significant improvements, whereas those using GEMINIS profiles show degradations, leading to an offset effect in the ENS. For weekly cycles, the ENS correlation remains unaltered in JFM and JAS, as improvements in some

- models balance degradations in others. For the diurnal cycle, results resemble those of  $PM_{10}$ , with a considerable increase incorrelation during OND (+0.15 in the ENS).
- Annual RMSE and bias scores for ENS remain largely unaffected by CAMS-REG-TEMPO for all four pollutants, although slight decreases are observed at higher concentration ranges, especially for PM<sub>10</sub>. While some seasonal differences emerge, these are minor compared to overall deviations from observations.







- The default temporal profiles differ across the 11 individual models, which partially explains the heterogeneous results observed. However, even among models using the same default profiles, contrasting responses to CAMS-REG-TEMPO are sometimes observed. For instance, NO<sub>2</sub> diurnal cycle correlations show opposite trends across models, which illustrates the complex interactions between temporal emission distributions and other physical and chemical processes such as the planetary boundary layer depth cycle. These findings align with previous air quality modelling intercomparisons exercises, where model spread persisted despite the use of common input parameters (e.g., Bessagnet et al., 2016).
- Overall, results indicate that the less the pollutant is directly linked to primary emissions, the lower is its sensitivity to changes in the emission temporal profiles. Improvements are particularly important for NO<sub>2</sub>, and to a lesser extent PM<sub>10</sub>, which are dominated by primary sources, while PM<sub>2.5</sub> and O<sub>3</sub> present a lower sensitivity due to a higher role of secondary formation and, in the case of O<sub>3</sub>, of the remote influences due to its higher lifetime.

All in all, the use of the CAMS-REG-TEMPO emission temporal profiles offers performance results encouraging enough to support their implementation in the operational CAMS multi-model ensemble production. As a matter of fact, several teams have already implemented them in their models (e.g., Ge et al., 2024; Menu et al., 2024; Soussé-Villa et al., 2025). Some of the profiles reported in the CAMS-REG-TEMPO dataset are based on meteorological parametrisations, such as the Heating Degree Days, which can significantly change between years. As discussed in detail in Guion et al., (2025), the implementation of online versions of these parametrisations within the CAMS models is recommended to improve the performance of models when used in forecasting mode. The CAMS-REG-TEMPO profiles used in this study can be obtained from Guevara et al. (2025). The profiles are categorised by temporal resolution, country, GNFR sector and pollutant, following the same nomenclature as the one used by the CAMS-REG emission inventory to facilitate their combination. While the present work focusses on quantifying the impact on the performance of the CAMS multi-model ensemble, the CAMS-REG-TEMPO profiles can also be adopted for other air quality modelling efforts beyond CAMS. This includes, for instance, the application of CAMS-REG-TEMPO for source apportionment and air quality planning studies (Thunis et al., 2018) and the assessment of the sensitivity of the associated modelling tools and results to changes in the anthropogenic emission temporal variability.

Future works will focus on evaluating the impact of CAMS-REG-TEMPO on other modelled species, including pollutants of emerging concern such as black carbon, NH<sub>3</sub> and individual NMVOC species, which may provide additional insights and allow identifying opportunities for improvement and further refinement of the proxies and parametrisations currently

considered to compute the profiles. We also plan to explore the development of new profiles for those activities for which we are still relying on data from the late nineties and that present significant contributions to primary emissions, namely NMVOC emissions from the use of solvent sector. Improvements will focus on investigating the inclusion of temperature-dependencies, as reported by recent studies such as Wu et al. (2024). The temporal redistribution of NMVOC emissions could have a substantial impact on individual modelled NMVOC species (e.g., toluene, xylene) and, to a lower extent, on modelled PM<sub>2.5</sub> 860 due to the important role of NMVOC from solvent use to the formation of fine secondary organic aerosols (SOA) (e.g. McDonald et al., 2018). For PM<sub>2.5</sub>, it is however important to note that the sensitivity of the modelling results to changes in the temporal profiles of NMVOC emissions will very much dependent on the SOA formation scheme that is implemented in the model. Currently, there are several models that include simplified SOA schemes, in which SOA precursor emissions from combustion sources are estimated using CO emissions as a proxy and therefore modelled SOA levels are not sensitive to 865 changes in primary NMVOC emissions (e.g., Pai et al., 2020). Despite being a precursor of O<sub>3</sub>, changes in the temporal allocation of primary NMVOC emissions may have a rather low impact on O<sub>3</sub> modelled concentrations. This hypothesis is based on the sensitivity results obtained from the present work, but also from other recent works that concluded that changes in the total amount or the speciation of anthropogenic NMVOC emissions translates into very limited changes of modelled O<sub>3</sub> concentrations (Petetin et al., 2023; Oliveira et al., 2025).


# 5 Code availability

The Python package evaluation of the modelling results can be downloaded from the following site: <a href="https://opensource.umr-cnrm.fr/projects/evaluation">https://opensource.umr-cnrm.fr/projects/evaluation</a> (last accessed: March 2025).

# 6 Data availability

The CAMS-REG-AP\_v4.2 gridded emission maps are accessible via <a href="https://doi.org/10.24380/0vzb-a387">https://doi.org/10.24380/0vzb-a387</a> (last accessed: March 2025). The CAMS-REG-TEMPO\_v3.2 temporal profiles are available at <a href="https://doi.org/10.5281/zenodo.15011343">https://doi.org/10.5281/zenodo.15011343</a> (last accessed: March 2025) (Guevara et al., 2025). Data on measurement stations from EEA can be downloaded at <a href="https://eeadmz1-downloads-webapp.azurewebsites.net/">https://eeadmz1-downloads-webapp.azurewebsites.net/</a> (last accessed: March 2025).

# 7 Author contribution

MG designed and drafted the overall manuscript, prepared the figures and coordinated all contributions. AC coordinated the air quality modelling intercomparison exercise. VP performed the evaluation of the air quality modelling results against observations. All co-authors contributed to the production of the air quality modelling results and to the discussion of the results.

# **8** Competing interests

The authors declare that they have no conflict of interest.

# 9 Acknowledgements



The research leading to these results has received funding from the Copernicus Atmosphere Monitoring Service (CAMS), which is implemented by the European Centre for Medium-Range Weather Forecasts (ECMWF) on behalf of the European Commission. AU acknowledges support of EU Horizon project CAMAERA (grant 101134927). BSC acknowledges support from the Department of Research and Universities of the Government of Catalonia via the Research Group Atmospheric Composition (code 2021 1090 SGR 01550). YP acknowledges the contribution of the EU Horizon project CATALYSE (grant 101057131). MS acknowledges the RCF project VFSP-WASE (grant 359421). FZJ ICE-3 gratefully acknowledges computing time on the supercomputer JURECA (Jülich Supercomputing Centre, 2021) at Forschungszentrum Jülich under grant no. cjicg21. The computing resources and the related technical support for MINNI are provided by CRESCO/ENEAGRID High Performance Computing infrastructure and its staff. CRESCO/ENEAGRID High Performance Computing infrastructure is funded by ENEA, the Italian National Agency for New Technologies, Energy and Sustainable Economic Development and by

Italian and European research programmes (see <a href="http://www.cresco.enea.it/english">http://www.cresco.enea.it/english</a>). We thank the two anonymous reviewers for their valuable comments and constructive suggestions on the manuscript.

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
