# Peer review of "Technical note: Sensitivity of the CAMS regional air quality modelling system to anthropogenic emission temporal variability"

_EGUsphere, 2025_

## Author Response (AR1)

The authors would like to thank the editor and the two anonymous reviewers for their prompt and thoughtful feedback on our manuscript. The answers to each specific comment can be found below in blue.

**Reviewer #1**

**General comments**

The manuscript presents a sensitivity analysis of the CAMS regional air quality modelling system to the temporal variability of anthropogenic emissions. This is a highly relevant and timely topic with valuable implications for the air quality modelling community. The study has potential to contribute significantly to the understanding of how emission timing affects model performance.

However, in its current form, the manuscript lacks sufficient depth in several key areas. The comparison of temporal profiles (monthly, weekly, hourly) would benefit from a more detailed, country-level analysis. For instance, do some countries show larger discrepancies across profiles? If so, are there data-related or methodological reasons behind these patterns? This level of detail would be particularly relevant, given that the model evaluation using observations is also conducted by country (the correlation coefficient). A clearer linkage between country-specific emission characteristics and model results would strengthen the overall interpretation.

We thank Reviewer's #1 for his/her time and for raising this point, as it is an important aspect of the analysis of the results.

Correlation coefficients between monthly, weekly and hourly emissions estimated using CAMS-REG-TEMPO versus TNO and CAMS-REG-TEMPO versus GENEMIS were estimated per pollutant and individual country to complement the analysis performed at the European scale. The results have been summarised in Figures S4 to S6 of the revised version of the manuscript. The largest discrepancies across profiles, pollutants and countries are highlighted and discussed in the revised version of the manuscript (Sections 3.1.1, 3.1.2 and 3.1.3).

A clearer linkage between differences in the temporally disaggregated emissions and data-related or methodological reasons behind them has also been included in the discussion of the results presented in Sections 3.1.1, 3.1.2 and 3.1.3. One example of these links is as follows:

"Concerning fertilizer emissions, the CAMS-REG-TEMPO country-dependent profiles are based on a mosaic of datasets including the regional European emission inventories reported for Denmark and Germany by Skjoth et al. (2011), for Poland by Werner et al. (2015), for the Netherlands, France and Belgium by Backes et al. (2016a) and from the global bottom-up MASAGE\_NH3 inventory for the rest of the countries (Paulot et al., 2014). In contrast, the TNO profile allocates the majority of NH3 emissions to March

(24.2% of total emissions), the peak being mainly driven by the profile proposed for fertilizer emissions. This profile is based on the work by Asman (1992), which considered information from the year 1989 in the Netherlands about the timing of manure spreading from different animal types as well as of application of synthetic fertilizers. Using CAMS-REG-TEMPO instead of TNO leads to a decrease in emissions by more than -50% during that month and an increase above 100% in summer (Fig. S1)." (lines 380 to 388 of the revised manuscript)

Additionally, the discussion of results is often limited to descriptive statements, with vague terms such as "similar" or "slight improvement" used without supporting metrics. Quantifying such terms (e.g., using statistical measures or percentage differences) would improve the clarity, objectivity, and scientific robustness of the analysis. The manuscript would also benefit from a more critical reflection on the causes of observed differences, particularly where the choice of temporal profiles leads to notable changes in model performance. In summary, while the manuscript addresses an important topic and presents promising results, more in-depth analysis, clearer quantification, and stronger interpretive discussion are needed to fully support the conclusions and maximize the manuscript contribution to the field.

We fully agree with Reviewer's #1 comment. We have included correlation coefficients, percentage differences and other supporting metrics to quantify the descriptive statements that appear in the text. Some of these supporting metric have not only been added to the text of the revised manuscript, but also summarised in new Tables, such as the correlation coefficients between monthly, weekly and hourly emissions estimated using CAMS-REG-TEMPO versus TNO and CAMS-REG-TEMPO versus GENEMIS per pollutant and at the EU27 plus UK and Norway level, which are now provided in Tables 4, 5 and 6, respectively.

We have quantified more than 30 descriptive statements, which allowed to improve the clarity, objectivity, and scientific robustness of the analysis. These quantifications have been included in the discussions of the results presented in Sections 3.1, 3.2 and 3.3 of the revised manuscript. Some examples of the quantifications and supporting metrics included in the revised version of the manuscript are as follows:

"The seasonality of NOx emissions is mainly dominated by the road transport (44.3% of total emissions) and industry sectors (energy and manufacturing, 33.5% of total emissions) (Fig. 1 and Fig. S1). The monthly cycles obtained with the three temporal profile databases present correlations of 0.67 (CAMS-REG-TEMPO versus TNO) and 0.79 (CAMS-REG-TEMPO versus GENEMIS) (Table 4)." (lines 339 to 342 of the revised manuscript)

"NH3 exhibits the largest differences in monthly emission distributions (Fig. 2), especially when comparing CAMS-REG-TEMPO and TNO profiles (**correlation coefficient of 0.39**).

CAMS-REG-TEMPO reports a distinct bi-modal seasonality, with a primary peak in April (15.3% of total emissions), mainly driven by fertilizer emissions (GNFR\_L), and a second lower-intensity peak in July (11.5% of total emissions), mainly linked to livestock emissions (GNFR\_K). (lines 377 to 380 of the revised manuscript)

"For SOx, the monthly emission cycles are largely dominated by the industry sector (81.5% of total emissions, Fig. 1). The seasonality obtained by each temporal profile database are largely correlated (0.9 and 0.89, Table 4)." (lines 363 to 364 of the revised manuscript)

We have also reviewed the discussion of results to clearly identify data-related or methodological reasons behind the improvements and dteriorations observed in the modelling results, making a clearer linkage between differences in the emission temporal profile databases characteristics and air quality modelling results. One example of these critical reflections on the causes of observed differences between modelling results are as follows:

"For O3, which exhibits a high seasonal variation, significant differences emerge between expA and expB during AMJ and JAS when both modelled and observed concentrations are at their maximum (Fig. 12). In AMJ, the median of the daily maximum concentration is lower with expB (94.9 μg/m3) compared to expA (96.2 μg/m3), which translates into a 28% increase of the bias between the ENS and observations. This bias increase is driven by the lower NOx emissions available to enhance O3 formation during April and May when using CAMS-REG-TEMPO instead of TNO or GENEMIS (approximately -10%, as indicated in Fig. S1). Conversely, during JAS, expB reports higher concentrations during July and the first half of August (104.4 and 106.9 and μg/m3 for expA and expB, respectively), reducing the bias when compared to observations, especially during the large-scale O3 pollution episodes occurred between the 23rd and 27th of July (bias reduction of -29.2%) and 2nd and 7th of August (bias reduction of -23.7%). We attribute this reduction in the biases to the larger amount of NOx emissions allocated to July and August when using CAMS-REG-TEMPO with respect to TNO and GENEMIS profiles (up to +8% according to Fig. S1)." (lines 1042 to 1052 of the revised manuscript)

**Specific comments**

Lines 100-102: Provide examples of the surrogate statistics considered, and specify the reference where the complete information can be found.

Examples of the spatial proxies considered and a reference where the complete information can be found have been added as follows:

"Some examples of spatial proxies include a road transport network with traffic intensities associated to each road link, which is used to distribute interurban traffic emissions, and a catalogue of industrial point sources with exact geographical

coordinates and emission strengths associated to each facility, which are used to distribute emissions from power plants and manufacturing industries. The summary of proxies used is provided in Kuenen et al. (2022)." (lines 156 to 164 of the revised manuscript)

Line 116: Could the authors clarify what is meant by "degree days" in this context? Is this simply referring to temperature, or is it a more specific metric (e.g., heating degree)?

We clarified in the text that we refer to heating degree days

**Lines 119-120: What type of data is used for energy and livestock emissions?**

A description of the type of data used for energy and livestock emission temporal profiles has been included as follows:

"(···) while energy sector profiles are derived from power plant fuel usage and load curves reported by Veldt (1992). Livestock emissions in TNO profiles are based on Skjøth et al. (2011), which developed a dynamic emission model that takes into account the effect of outdoor temperatures in NH3 emissions from animal houses or manure storages" (lines 181 to 186 of the revised manuscript)

**Line 121: What do you mean by "identical"? Which sectors have different hourly, sector-dependent profiles?**

The sentence was not well formulated. We modified it to clearly indicate that the hourly temporal profiles proposed by the GENEMIS and TNO databases are the same:

"Both GENEMIS and TNO report the same hourly sector-dependent profiles" (lines 185 to 186 of the revised manuscript)

**Lines 146-147: Did the authors classify the automatic traffic stations as urban or rural using the GHSL dataset?**

Yes, We added a sentence describing the procedure we considered:

"For that, the original 1km  $\times$  1km GHSL raster was remapped onto the CAMS-REG-AP grid (0.1°  $\times$  0.05°) following a majority resampling method, in which each destination grid cell was assigned with the GHSL classification that had a higher number of occurrences within that grid cell." (lines 225 to 227 of the revised manuscript)

**Lines 202-204: How are the pollutant-specific thresholds defined? The authors should support these values with appropriate references.**

The pollutant-specific thresholds were defined on the basis of probability distributions of concentrations measured in Europe over 8 years, to discard potentially spurious values outside the distribution. These thresholds were defined qualitatively to work on data sets that were not always (or incompletely) "validated", for practical reasons as a preventive

measure, in order to avoid a too strong impact of extreme values on certain forecast scores. This clarification has been added to the revised version of the manuscript

**Line 119: Why was the year 2018 selected for the simulations?**

The intercomparison exercise presented in the manuscript started in 2022. Due to lockdown episodes in most European countries, 2020 (and partially 2021) were not representative years for testing the different temporal profiles of emissions.

The year 2018 was chosen by convenience due to a previous modeling exercise involving several models of the Ensemble (CAMS-61, "Evaluation and development of regional air quality modelling and data assimilation aspects"; Timmermans, 2021). The experimental configuration was therefore already partially set for 2018.

Furthermore, 2018 was an interesting year from a scientific point of view due to the occurrence of summer episodes of ozone air pollution linked to heat waves and intense summer droughts in Europe (e.g. Pope et al., 2023).

This information has been added to the revised version of the manuscript as follows:

"The year 2018 was chosen by convenience due to a previous modeling exercise involving several models of the CAMS ensemble (Timmermans, 2021). Furthermore, 2018 was an interesting year from a scientific point of view due to the occurrence of summer episodes of ozone air pollution linked to heat waves and intense summer droughts in Europe (e.g., Pope et al., 2023)." (lines 301 to 304 of the revised manuscript)

Pope, R. J., Kerridge, B. J., Chipperfield, M. P., Siddans, R., Latter, B. G., Ventress, L. J., Pimlott, M. A., Feng, W., Comyn-Platt, E., Hayman, G. D., Arnold, S. R., and Graham, A. M.: Investigation of the summer 2018 European ozone air pollution episodes using novel satellite data and modelling, Atmos. Chem. Phys., 23, 13235–13253, https://doi.org/10.5194/acp-23-13235-2023, 2023.

Timmermans, R.. Evaluation and development of regional air quality modelling and data assimilation aspects - Highlights from CAMS-61, Available at: https://atmosphere.copernicus.eu/sites/default/files/custom-uploads/CAMS-5thGA/day1/Timmermans%20R\_TNO\_European%20air%20quality.pdf, 2021.

**Line 223: Why was the 2017 inventory used? The authors should justify this choice and explain whether it has any implications for the objectives of the paper.**

At the time of starting the simulations, 2017 was the most recent year for which anthropogenic CAMS-REG emissions were available. According to CEIP (2025), between 2017 and 2018 (the study year for the simulations) primary anthropogenic emissions presented very limited changes at the EU27 level (e.g., -4.3% for NOx and 2% for PM2.5). These relative changes are lower than the uncertainty of the officially reported emission inventories (Guion et al., 2025).

CEIP: EMEP Centre on Emissions Inventories and Projections. Officially reported emission data, https://www.ceip.at/webdab-emission-database/reported-emissiondata (last access: July 2025), 2025.

Guion, A., Couvidat, F., Guevara, M., and Colette, A.: Country- and species-dependent parameters for the heating degree day method to distribute NOx and PM emissions from residential heating in the EU 27: application to air quality modelling and multi-year emission projections, Atmos. Chem. Phys., 25, 2807–2827, https://doi.org/10.5194/acp-25-2807-2025, 2025.

**Line 244: What do you mean by "similar"? This should be quantified, for example, by using a correlation coefficient.**

To improve the clarity of the discussion of the emission results, we have included three new tables (Tables 4 to 6 of the revised manuscript) summarising the correlation coefficients between monthly, weekly and hourly emissions estimated using CAMS-REG-TEMPO versus TNO and CAMS-REG-TEMPO versus GENEMIS per pollutant. These correlation coefficients are used to quantify the descriptive statements, following reviewer's recommendations. Concerning the specific case highlighted by the reviewers, the quantification has been introduced as follows:

"The monthly cycles obtained with the three temporal profile databases present correlations of 0.67 (CAMS-REG-TEMPO versus TNO) and 0.79 (CAMS-REG-TEMPO versus GENEMIS) (Table 4), with differences ranging between -10 and 10% depending on the month (Fig. S1)." (lines 340 to 342 of the revised manuscript)

These correlation coefficients are referenced in other parts of the discussion of the results presented in Sections 3.1.1, 3.1.2 and 3.1.3 of the revised manuscript, to improve the clarity, objectivity, and scientific robustness of the analysis.

**Lines 250-251: Do TNO and GENEMIS provide static profiles, or do these datasets not offer year-specific profiles based on meteorological data?**

The TNO and GENEMIS profiles do not offer year-specific profiles based on meteorological data. We have clarified this point as follows:

"These profiles result in a stronger contrast between cold and warm months, leading to higher emissions during colder periods compared to the profiles used in TNO and GENEMIS, which do not offer year-specific weight factors based on meteorological data." (lines 348 to 349 of the revised manuscript)

**Lines 257-258: Why is there this discrepancy? What is the difference between the methodologies used to obtain these results?**

The following clarification has been added:

"For CAMS-REG-TEMPO, the seasonality of these emissions is linked to agricultural waste burning emissions and is derived from Klimont et al. (2017), which considered the timing and location of active fires on agricultural land in the Global Fire Emissions Database (GFEDv3.1). In GENEMIS, the profile proposed for NMVOC agricultural activities is based on statistical data on sales and application of agricultural pesticides or other agrochemicals (Friedrich and Reis, 2004)." (lines 358 to 361 of the revised manuscript)

Line 260: What are the winter months? In Europe, the winter months are December, January, and February, but this is not clearly stated in the document.

Winter months are December, January, and February. We added a remark in the text to clarify this aspect:

"Compared to winter (i.e., December, January and February), the drop in industrial emissions  $(\cdots)$ " (lines 364 to 365 of the revised manuscript)

Line 268: Is the lower-intensity peak in June or July? Please verify.

The lower-intensity peak is occurring in July. We have corrected it in the revised text

Line 288-289: What type of information is used in the TNO profiles for the residential and commercial combustion sector?

The TNO profile for the residential and commercial combustion sector is based on on fuel use information from small consumers as reported by Veldt et al. (1992). This information has been added to the revised version of the manuscript as follows:

"The CAMS-REG-TEMPO and GENEMIS profiles are similar, as both consider the impact of meteorology (i.e., temperature-driven variations in heating demand), while for TNO the profile is based on fuel use information from small consumers (Veldt et al., 1992)." (lines 417 to 419 of the revised manuscript)

Line 303: How large is the "significantly larger drop"? Please provide a quantification.

A quantification has been provided as follows:

"The NOx weekly cycle (Fig. 3) obtained with CAMS-REG-TEMPO presents a significantly larger drop of emissions between weekdays and weekends (-38%) compared to TNO (-22%) and GENEMIS (-15%)." (lines 445 to 446 of the revised manuscript)

Line 308: What do you mean by 'lower extent'? This type of statement should be quantified or clarified for better precision. Please ensure this issue is addressed consistently throughout the document (e.g., nearly identical, almost identical, identical, similar)

The statement has been quantified as follows:

"These discrepancies are mainly driven by differences in the weekly profiles for road transport, which present a 44% contribution to total NOx emissions at EU27 plus UK and Norway scale, and, to a lower extent, for off-road transport (included in the "Others" category), which contribution is of 10%" (lines 449 to 452 of the revised manuscript)

As indicated in a previou comments, qualitative statements similar to the one highlighted in this comment have been quantified throughout the manuscript to improve the clarity of the analysis and discussions of the results.

Lines 303-330: The authors should try justify the results based on the datasets used for each European temporal profile. For example, lines 355-360, what type of data is used by TNO for road transport? What type of data is used by CAMS-REG-TEMP and TNO for off-road transport?

Clarifications on the type and sources of information considered for the development of the TNO, GENEMIS and CAMS-REG-TEMPO profiles have been added in sections 3.1.1, 3.1.2 and 3.1.3. Clarifications have been included not only for the road transport and offroad transport sectors, but also for the public power, residential combustion, agriculture and use of solvent sectors.

**Lines 403-404: This statement is not clear.**

We have reformulated the statement to make it more clear:

"The results for each individual model are provided in the supplementary material (Fig. S7). Please note that due to technical issues during the simulations, the modeled concentrations of DEHM (MATCH) NO2 and O3 (PM10 and PM2.5) were excluded from the comparative analysis and are therefore not available in the supplementary material" (lines 634 to 636 of the revised manuscript)

**Lines 411-412: What could be the cause of this issue? Could it be related to the VOC emission profiles?**

As later explained in Section 3.1.2, this result illustrates the complexity of the  $O_3$  cycle, which exhibits non-linear relationships with its main precursors, NOx and VOCs. All the individual CAMS models except EMEP, which presents an improvement of the weekly  $O_3$  correlation for all four seasons, show the same issue. One explanation could be the potential influence of the online biogenic NMVOC and soil  $NO_x$  emission parametrisations considered in each CAMS model, as described in Colette et al. (2024). Concerning the role of anthropogenic NMVOC emissions, recent works have shown that changes in the amount or the speciation of anthropogenic NMVOC emissions translated into very limited or almost negligible changes of modelled  $O_3$  concentrations during the summer season, when  $O_3$  levels are at their maximum (Petetin et al., 2023; Oliveira et al., 2025). As shown in Fig.8, for  $O_3$  there is a clear spatial pattern in the results during JAS, the scores being slightly improved in Central Europe and slightly deteriorated in Western

and Eastern Europe. One aspect that is interesting to highlight about the slight deterioration of the scores in Western Europe is that it mainly affects rural areas (as opposed to urban areas). This is clearly visible for France and Spain, where we can see that in stations located in the respective capitals (Paris and Madrid) and other urban areas (Marseille, Barcelona) correlations are increasing, while in rural regions scores are being deteriorated. These results highlight the added value of the new CAMS-REG-TEMPO profiles for areas with high NOx emissions, particularly the profiles proposed for the road transport sector, which is the main dominant source of NOx emissions in urban areas. The deterioration observed in rural areas is aligned with the hypothesis that natural emissions, which are dominant in these regions, may be playing a role in the obtained results.

Colette, A., Collin, G., Besson, F., Blot, E., Guidard, V., Meleux, F., Royer, A., Petiot, V., Miller, C., Fermond, O., Jeant, A., Adani, M., Arteta, J., Benedictow, A., Bergström, R., Bowdalo, D., Brandt, J., Briganti, G., Carvalho, A. C., Christensen, J. H., Couvidat, F., D'Elia, I., D'Isidoro, M., Denier van der Gon, H., Descombes, G., Di Tomaso, E., Douros, J., Escribano, J., Eskes, H., Fagerli, H., Fatahi, Y., Flemming, J., Friese, E., Frohn, L., Gauss, M., Geels, C., Guarnieri, G., Guevara, M., Guion, A., Guth, J., Hänninen, R., Hansen, K., Im, U., Janssen, R., Jeoffrion, M., Joly, M., Jones, L., Jorba, O., Kadantsev, E., Kahnert, M., Kaminski, J. W., Kouznetsov, R., Kranenburg, R., Kuenen, J., Lange, A. C., Langner, J., Lannuque, V., Macchia, F., Manders, A., Mircea, M., Nyiri, A., Olid, M., Pérez García-Pando, C., Palamarchuk, Y., Piersanti, A., Raux, B., Razinger, M., Robertson, L., Segers, A., Schaap, M., Siljamo, P., Simpson, D., Sofiev, M., Stangel, A., Struzewska, J., Tena, C., Timmermans, R., Tsikerdekis, T., Tsyro, S., Tyuryakov, S., Ung, A., Uppstu, A., Valdebenito, A., van Velthoven, P., Vitali, L., Ye, Z., Peuch, V.-H., and Rouïl, L.: Copernicus Atmosphere Monitoring Service – Regional Air Quality Production System v1.0, EGUsphere [preprint], https://doi.org/10.5194/egusphere-2024-3744, 2024.

Petetin, H., Guevara, M., Garatachea, R., López, F., Oliveira, K., Enciso, S., Jorba, O., Querol, X., Massagué, J., Alastuey, A. and Pérez García-Pando, C.: Assessing ozone abatement scenarios in the framework of the Spanish ozone mitigation plan. Sci. Total Environ., 902 (2023), Article 165380, 10.1016/J.SCITOTENV.2023.165380.

Oliveira, K., Guevara, M., Kuenen, J., Jorba, O., Pérez García-Pando, C. and Denier van der Gon, H.: Enhancing anthropogenic NMVOC emission speciation for European air quality modelling, Environmental Pollution, 382, 126510, <a href="https://doi.org/10.1016/j.envpol.2025.126510">https://doi.org/10.1016/j.envpol.2025.126510</a>, 2025.

**Line 415-416: Which model is the exception? Is there any explanation for this?**

This statement has been clarified as follows:

"The improvements in NO2 weekly correlations are consistent across all models except for MATCH, which largely increases the correlations during AMJ (0.19) and JAS (0.49) but also shows slight degradations during JFM (-0.09) and OND (-0.08)." (lines 647 to 648 of the revised manuscript)

The degradations reported by MATCH for JFM and OND are much lower than the improvements observed during AMJ and JAS.

Lines 420-421: What could be the reason for this? How does the monthly PM10 temporal profile used by CAMS-REG-TEMP differ from the others?

The monthly correlations reported by the ENS in expA for  $O_3$ ,  $NO_2$  and  $PM_{2.5}$  are already very high (0.95, 0.83 and 0.82, respectively), while  $PM_{10}$  presents the lowest correlation (0.68), giving more room for improvement. We have added this clarification in the revised version of the manuscript.

Figure 10: It is unclear why the authors chose to present NO2 weekly cycle concentrations specifically for JFM and JAS. Do these periods represent the highest and lowest values? If so, it would be helpful to clarify this in the figure or caption to guide the reader (the same comment for the other pollutants).

JFM and JAS periods were selected because, on the one hand, they represent winter-like and summer-like conditions and, on the other hand, they also represent the highets and lowest values of the year. This clarification has been added on the caption of Figure 10 as follows:

"The JFM and JAS periods were selected because they represent winter-like and summer-like conditions as well as the highest and lowest values of the year." (lines 775 to 777 of the revised manuscript)

Similar clarifications have been added in the captions of Figures 11, 13, 14, 16, 17, 19 and 20 Moreover, the weekly and diurnal cycles for the periods not presented in the manuscript were added in the revised version of the supplementary material (Figures S8 and S9).

Figure 10: It would be helpful to also include the results from the CAMS European multi-model ensemble air quality modelling system (not only by model) for comparison (the same suggestion applies to the other pollutants). This would allow for a more complete assessment of model performance.

The results from the CAMS European multi-model ensemble air quality modelling system (ENS) are already included in Figure 10 and all the other Figures (11 to 20). The results are reported under the category ENS (first two columns in the X-axis).

**Lines 524-525: What could be the reason for this reduction in accuracy?**

The lower PM10 monthly correlation in the CHIMERE and EMEP models is related to a less accurate reproduction of the observed  $PM_{10}$  level increases between January and February (CHIMERE) and September and October (EMEP) when moving from GENEMIS (expA) to CAMS-REG-TEMPO profiles (expB). For the first case, the degradation could be linked to the fact that  $NH_3$  emissions, which largely contribute to the formation of

secondary inorganic aerosols during cold months (e.g. Backes et al., 2016b; Clappier et al., 2021), remain constant between January and February when using CAMS-REG-TEMPO (+0.5% increase), while a large increase is observed when considering the GENEMIS profiles (+14%), as reported in Fig. 2. For the second case, the reduction in accuracy could be related to the lower increase in primary PM10 emissions between September and October reported by CAMS-REG-TEMPO (32%) when compared to GENEMIS (38%), combined with the -13% decrease (9% increase) of total NH3 emissions reported by CAMS-REG-TEMPO (GENEMIS) for the same period.

This clarification has been added to the revised version of the manuscript.

Backes, A.M., Aulinger, A., Bieser, J., Matthias, V., Quante, M.: Ammonia emissions in Europe, part II: How ammonia emission abatement strategies affect secondary aerosols, Atmospheric Environment, 126, 153-161, 2016b.

Clappier, A, Thunis, P., Beekmann, M., Putaud, J.P. and de Meij, A.: Impact of SOx, NOx and NH3 emission reductions on PM2.5 concentrations across Europe: Hints for future measure development. Environ Int., 156:106699. doi: 10.1016/j.envint.2021.106699, 2021.

Lines 521-522: Information such as "8 out of 11" could be added for the remaining pollutants and temporal profiles. This type of detail would help support the authors' statements (e.g., "slight improvement").

Following the reviewer's suggestion, we have added this information in the discussion of the results presented in Sections 3.2.1 to 3.2.4 of the revised version of the manuscript.

Section 3: The authors need to provide specific values; for example, what does "slight decrease" mean? Why did the authors obtain these particular results? Why were only O3 and PM2.5 results presented? Why are only two seasons represented for each pollutant? A more thorough explanation would enhance the clarity and robustness of the analysis.

Specific values were added to quantify the impact of the CAMS-REG-TEMPO on the annual spatial median of bias and RMSE for  $NO_2$ ,  $O_3$ ,  $PM_{10}$  and  $PM_{2.5}$ . The information provided in the revised version of the manuscrit is the following:

"Overall, the statistics hardly vary between experiments, although slight decreases in both bias and RMSE are observed at higher concentration ranges when comparing expB to expA for O3 (bias and RMSE reductions of -2.4% and -1.4% for concentrations ranging from 110 to 130  $\mu$ g/m3 and of -1.1% and -0.8% for concentrations ranging from 130 to 150  $\mu$ g/m3) and PM10 (bias and RMSE reductions of -2.1% and -1.3% for concentrations equal or larger than 60  $\mu$ g/m3). Concerning NO2, the larger improvements are observed for concentrations raging between 40 and 60  $\mu$ g/m3 (bias and RMSE reductions of -3.1% and -2.4%), while in the case of PM2.5 the reduction of the bias and RMSE is mainly occurring at low concentration ranges (bias and RMSE reductions of -11.9% and -0.5%

for concentrations ranging from 5 to 10  $\mu g/m3$ )." (lines 1016 to 1022 of the revised manuscript)

Concerning the analysis of observed and modelled daily maximum and daily mean concentrations of  $O_3$  and  $PM_{2.5}$ , we selected the seasons where concentrations are at their maximum (AMJ and JAS for  $O_3$ ) and that represent both winter and summer-like conditions (JFM and JAS for  $PM_{2.5}$ ). Results for other seasons have been added to the supplementary material and referenced in the main manuscript.

"Unlike annual averages, concentrations can vary significantly between seasons. To complement this analysis, Fig. 22 shows the spatial median of the observed and modelled (ENS, expA and expB) daily maximum concentration of  $O_3$  and daily mean concentration of  $NO_2$  and  $PM_{2.5}$  for selected seasons. The selected seasons represent winter-like and summer-like conditions as well as the highest and lowest concentration values of the year. Results for the remaining seasons and for  $PM_{10}$ , which conclusions are almost identical to the ones obtained for  $PM_{2.5}$ , are reported in the Supplementary material (Figure S10)." (lines 1036 to 1040 of the revised manuscript)

Moreover, we have also included results for  $NO_2$  and selected seasons (JFM and OND) in the revised version of Figure 22. The conclusions obtained from the  $PM_{10}$  are almost identical to the ones derived from  $PM_{2.5}$ , and therefore have been added to the supplementary material.

The whole Section has been improved by including more detailed and quantified explanations of all the results to enhance the clarity and robustness of the analysis.

**Technical corrections**

Line 79: The final punctuation mark is missing.

**Added**

Table 1: The abbreviations should be defined. The authors can include this information at the bottom of the table.

Definitions of abbreviations have been added to the table. Note that The CHIMERE chemistry-transport model is written in capital letters (e.g., Menut et al. 2013 and 2021) but is not an abbreviation or acronym.

Menut, L., Bessagnet, B., Khvorostyanov, D., Beekmann, M., Blond, N., Colette, A., Coll, I., Curci, G., Foret, G., Hodzic, A., Mailler, S., Meleux, F., Monge, J.-L., Pison, I., Siour, G., Turquety, S., Valari, M., Vautard, R., and Vivanco, M. G.: CHIMERE 2013: a model for regional atmospheric composition modelling, Geosci. Model Dev., 6, 981–1028, https://doi.org/10.5194/gmd-6-981-2013, 2013.

Menut, L., Bessagnet, B., Briant, R., Cholakian, A., Couvidat, F., Mailler, S., Pennel, R., Siour, G., Tuccella, P., Turquety, S., and Valari, M.: The CHIMERE v2020r1 online chemistry-transport model, Geosci. Model Dev., 14, 6781–6811, https://doi.org/10.5194/gmd-14-6781-2021, 2021.

Line 101: Replace "x" (the letter) with "x" (the multiplication symbol). Please ensure this formatting is consistent throughout the document.

Symbol replaced and consistency ensured throughout the document

Line 117: Please verify whether "TNO profiles" should be referred to as "TNO\_MACC-III profiles".

TNO\_MACC-III reffers to a specific version of the European regional emission inventory developed under the MACC III program, the last of the pre-operational stages in the development of the Copernicus Atmosphere Service (CAMS). While the original temporal profiles were developed under MACC, these have been modified under CAMS phase 1 (2015-2020) by, e.g., moving from SNAP-dependent profiles to GNFR-dependent profiles. We therefore reffer to them as TNO profiles.

Line 131: The pollutant names should be defined, and the "10" and "2.5" in  $PM_{10}$  and  $PM_{2.5}$  should be properly formatted as subscripts. Please ensure this formatting is consistent throughout the document.

We addded the definitions of the pollutant names and ensured a correct formatting for  $PM_{10}$  and  $PM_{2-5}$  throughout the document

Line 135: Please verify the sentence.

We corrected the sentence

Line 202: I suggest replacing the word "aberrant" with "outliers," for example.

Replaced

Line 244: Include also the Fig. S1 ("Fig. 1 and Fig. S1").

Included

Figure 2: There is a strange black line in the NH3 graphs. It seems that the GNFR lines are too close to each other. I suggest removing the black border from the stacked lines plots to improve readability (e.g., edgecolor='none').

Following the reviewer's recommendation, we have modified Figures 1 to 6 by removing the black border from the stacked line plots

Line 342: Should "GENEMIS-Menutetal2012" be replaced by "GENEMIS"? Please ensure consistency throughout the document. Please check also the Figure S3.

As indicated in the introduction of Section 3.1, for hourly emission cycles we excluded GENEMIS from the intercomparison analysis, as they report the exact same sector-dependent hourly profiles as TNO. Instead, an additional dataset was included in the comparison: the default hourly temporal factors used in EMEP and CHIMERE, which combine GENEMIS hourly profiles (identical to TNO profiles) with the country-dependent road transport profiles from Menut et al., (2012). We refer to this dataset as GENEMIS-Menuetal2012. To avoid confusions, we have added an additional clarification at the beginning of Section 3.1.3 of the revised manuscript.

We have also corrected the caption and titles of barplots of Figure S3

Figure 7: The sentence "Boxes highlighted in green/orange/grey indicate" should be revised -there are no orange boxes in the figure.

We replaced "orange" with "salmon"

Figure 9: The unit for the correlation coefficient should be provided (i.e., (-)). Additionally, the "3" should be formatted as a superscript. It would also be helpful to add the month abbreviations on the x-axis, as shown in Figure 2. The same comment applies to the remaining figures.

Figures 9 to 22 have been modified following the reviewers comments.

Line 500: remove the parenthesis "(".

Removed

Line 533: GEM-AQ or GEMAQ?

The name of the model is GEM-AQ. We have modified Figures 9 to 20, since in those we were referring to it as GEMAQ.

**Reviewer #2**

The manuscript submitted by the authors presents a thorough and diligent effort comparing different temporal disaggregation methods for emissions against traditionally used profiles. The study is well-structured and addresses one of the current key challenges in air quality modeling: the temporal allocation of emission inventories and the sensitivity of Chemical Transport Models (CTMs) to such changes. The work includes advancements that may be highly relevant for future modeling efforts. A clear improvement in simulated NO2 concentrations is observed due to the introduced refinements. However, some limitations remain, which pose ongoing challenges for the air quality modeling community in Europe. I believe the manuscript does not require major revisions.

We thank Reviewer #2 for taking the time to review the manuscript and for providing valuable comments.

That said, I would have appreciated the inclusion of a set of recommendations or at least some hypotheses for the future application of the proposed temporal profiles. It is implied that the authors would recommend adopting this new temporal allocation, but this could be stated more explicitly.

The following statement has been added to state more explicitly the recommendation of adopting these new temporal profiles and future potential applications beyond CAMS:

"The CAMS-REG-TEMPO profiles used in this study can be obtained from Guevara et al. (2025). The profiles are categorised by temporal resolution, country, GNFR sector and pollutant, following the same nomenclature as the one used by the CAMS-REG emission inventory to facilitate their combination. While the present work focusses on quantifying the impact on the performance of the CAMS multi-model ensemble, the CAMS-REG-TEMPO profiles can also be adopted for other air quality modelling efforts beyond CAMS. This includes, for instance, the application of CAMS-REG-TEMPO for source apportionment and air quality planning studies (Thunis et al., 2018) and the assessment of the sensitivity of the associated modelling tools and results to changes in the anthropogenic emission temporal variability." (lines 1157 to 1163 of the revised manuscript)

One particular result is the spatial behavior of ozone, which shows deterioration in Western and Eastern Europe, but not in Central Europe. The authors attribute this to the inherent difficulties in modeling ozone and highlight once again the nonlinear relationship between O3 and its precursors. Still, there appears to be a discernible pattern that might benefit from further exploration. I suggest the authors propose a hypothesis to explain this phenomenon.

Firstly, it is important to note that the map showing the ENS correlation differences (expB – expA) at the station level for  $O_3$  (Fig.8, top-right) is using a range of absolute values quite low (between -0.03 and +0.03) when compared to the ones used for the other species in the same figure (-0.1 and +0.1 for  $NO_2$  and  $PM_{10}$  and -0.08 and +0.08 for  $PM_{2.5}$ ). We believe putting all the maps on the same scale might be less interesting as results, so that's why we opted for using pollutant-dependent ranges.

Having said that, the reviewer correctly points out that for  $O_3$  there is a clear spatial pattern in the results, the scores being slightly improved in Central Europe and slightly deteriorated in Western and Eastern Europe. One aspect that is interesting to highlight about the slight deterioration of the scores in Western Europe is that it mainly affects rural areas (as opposed to urban areas). This is clearly visible for France and Spain, where we can see that in stations located in the respective capitals (Paris and Madrid) and other urban areas (Marseille, Barcelona) correlations are increasing, while in rural regions scores are being deteriorated. These results highlight the added value of the new CAMS-REG-TEMPO profiles for areas with high  $NO_x$  emissions, particularly the profiles proposed for the road transport sector, which is the main dominant source of  $NO_x$  emissions in urban areas. Since the deterioration is mainly occurring in rural areas, one hipothesis to explain these results could be the potential influence of the online biogenic NMVOC and soil  $NO_x$  emission parametrisations considered in each CAMS model, as described in Colette et al. (2024).

This discussion has been added to the revised version of the manuscript

Similarly, a brief discussion or hypothesis regarding the behavior of PM and the observed shift would add value. While it is helpful that the authors identify both improvements and deteriorations resulting from the redistribution of emissions, articulating hypotheses and providing recommendations for future modeling would strengthen the scientific impact of the work.

As shown in Figures 17 and 20 of the manuscript, the ENS modelling results present a problem in the reproduction of the observed PM10 and PM2.5 morning peaks, with a shift of approximately two hours between the modelled and measured peaks being observed both in the expA and expB ENS results. This PM peak shift problem is frequent and known for several years. As indicated by Schaap et al. (2011), this issue could be related to limitations in the reproduction of the diurnal cycles of inorganic aerosols (e.g., nitrate, sulphate, ammonium, nitric acid and ammonia). Another relevant aspect that could be driven the shift between modelling results and observations are transport and/or chemical reaction pathways relevant to the formation of secondary organic aerosols that are not adequately included in chemical transport models' input or formulation, as reported by Mircea et al., (2019). Other aspects that could explain the limitations of the mdelling results could be the representation of dynamic processes (e.g., diffusion, advection) at fine scale in cities with chemical transport models running at ~10km

resolution and the dynamics of the height of the boundary layer, which can be difficult to simulate in regions with complex topography.

These hypothesis have been included in the revised version of the manuscript as follows:

"This PM peak shift problem is frequent and known for several years. As indicated by Schaap et al. (2011), this issue could be related to limitations in the reproduction of the diurnal cycles of inorganic aerosols (e.g., nitrate, sulphate, ammonium, nitric acid and ammonia). Another aspect that could be driven the shift between PM modelling results and observations are transport and/or chemical reaction pathways relevant to the formation of secondary organic aerosols that are not adequately included in chemical transport models' input or formulation, as reported by Mircea et al., (2019). Other aspects that could explain the limitations of the modelling results could be the representation of dynamic processes and the development of the boundary layer, which can be difficult to simulate in regions with complex topography with chemical transport models running at ~10km resolution." (lines 898 to 910 of the revised manuscript)

Schaap, M., Otjes, R. P., and Weijers, E. P.: Illustrating the benefit of using hourly monitoring data on secondary inorganic aerosol and its precursors for model evaluation, Atmos. Chem. Phys., 11, 11041–11053, https://doi.org/10.5194/acp-11-11041-2011, 2011.

Mircea, M., Bessagnet, B., D'Isidoro, M., Pirovano, G., Aksoyoglu, S., Ciarelli, G., Tsyro, S., Manders, A., Bieser, J., Stern, R., García Vivanco, M., Cuvelier, C., Aas, W., Prévôt, A., Aulinger, A., Briganti, G., Calori, G., Cappelletti, A., Colette, A., Couvidat, F., Fagerli, H., Finardi, S., Kranenburg, R., Rouïl, L., Silibello, C., Spindler, G., Poulain, L., Herrmann, H., Jimenez, J.L., Day, D.A., Tiitta, P and Carbone, S.: EURODELTA III exercise: An evaluation of air quality models' capacity to reproduce the carbonaceous aerosol, Atmospheric Environment: X, 2, 100018, https://doi.org/10.1016/j.aeaoa.2019.100018, 2019.

It would also be beneficial to briefly mention the speciation approach used for emissions. While I assume the speciation follows standard practice.

Information on the speciation approach was added in Section 2.2 as follows:

"NMVOC and PM emissions are speciated using the sector- and country-dependent speciation profiles provided in CAMS-REG, which allow break downing the total NMVOC to the 25 Global Emission InitiAtive (GEIA) species (Schultz et al., 2007) and the total PM emissions to primary organic carbon, elemental carbon, sulphates, sodium and others. Each individual CAMS modelling team performs a remapping of the 25 GEIA NMVOC species and individual PM component to the species used in their corresponding gas phase and aerosol chemical mechanisms" (lines 160 to 164 of the revised manuscript)

Another point worth a brief mention in the manuscript is the selection of the year 2018. While it is understandable that 2018 can be considered a representative and valid year for the analysis, a short explanation of why this particular year was chosen would add helpful context.

The intercomparison exercise presented in the manuscript started in 2022. Due to lockdown episodes in most European countries, 2020 (and partially 2021) were not representative years for testing the different temporal profiles of emissions.

The year 2018 was chosen by convenience due to a previous modeling exercise involving several models of the Ensemble (CAMS-61, "Evaluation and development of regional air quality modelling and data assimilation aspects"; Timmermans, 2021). The experimental configuration was therefore already partially set for 2018.

Furthermore, 2018 was an interesting year from a scientific point of view due to the occurrence of summer episodes of ozone air pollution linked to heat waves and intense summer droughts in Europe (e.g. Pope et al., 2023).

This information has been added to the revised version of the manuscript as follows:

"The year 2018 was chosen by convenience due to a previous modeling exercise involving several models of the CAMS ensemble (Timmermans, 2021). Furthermore, 2018 was an interesting year from a scientific point of view due to the occurrence of summer episodes of ozone air pollution linked to heat waves and intense summer droughts in Europe (e.g., Pope et al., 2023)." (lines 301 to 304 of the revised manuscript)

Pope, R. J., Kerridge, B. J., Chipperfield, M. P., Siddans, R., Latter, B. G., Ventress, L. J., Pimlott, M. A., Feng, W., Comyn-Platt, E., Hayman, G. D., Arnold, S. R., and Graham, A. M.: Investigation of the summer 2018 European ozone air pollution episodes using novel satellite data and modelling, Atmos. Chem. Phys., 23, 13235–13253, https://doi.org/10.5194/acp-23-13235-2023, 2023.

Timmermans, R.: Evaluation and development of regional air quality modelling and data assimilation aspects - Highlights from CAMS-61, Available at: <a href="https://atmosphere.copernicus.eu/sites/default/files/custom-uploads/CAMS-5thGA/day1/Timmermans%20R\_TNO\_European%20air%20quality.pdf">https://atmosphere.copernicus.eu/sites/default/files/custom-uploads/CAMS-5thGA/day1/Timmermans%20R\_TNO\_European%20air%20quality.pdf</a>, 2021.

**Regarding the selection of air quality monitoring stations, is there a reference or link to the specific stations used that could be consulted (list of stations)?**

We have included a list of air quality monitoring stations used for the evaluation of the modelling results in the Supplementary material (Table S2). For each stations we provide information on the code, type (bac: background), area(sub: suburban, urb: urban, rur: rural), geographical coordinates (lat: latitude, lon: longitude) and pollutants monitored. The table is references in the revised version of the manuscript as follows:

"The complete list of air quality monitoring stations used for the evaluation of the modelling results is provided in Table S2." (lines 284 to 285 of the revised manuscript)

The ozone curve seems to show a certain regional bias—perhaps linked to traffic patterns? I am not familiar with overall averaged curves.

We attribute the positive bias of  $O_3$  nightime levels reported in Fig. 14 to the negative bias of the modelled  $NO_x$  levels (Fig. 11), which lead to an underestimation of  $O_3$  loss via NO titration. The  $O_3$  nighttime overestimation is a common feature of air quality models and has been extensevily discussed in previous works (e.g., Bessagnet et al., 2016; Sharma et al., 2017; Pay et al., 2019). This clarification has been added to the revised version of the manuscript.

Finally, the authors rightly highlight the uncertainties associated with NMVOC emissions, particularly given that a significant share comes from solvent use, and that this profile remains largely unchanged. As an additional comment (not necessarily for inclusion in the manuscript), I am curious whether the authors believe that the temporal redistribution in these emissions could have a more substantial impact on concentrations, and on which pollutants in particular. This could be an important consideration for future modeling work, especially as speciation and analysis of NMVOCs become increasingly relevant for air quality.

Our hypothesis is that the temporal redistribution of NMVOC emissions could have a sustantial impact on individual modelled NMVOC species (e.g., benzene, toluene, xylene, which are the most commonly monitored species) and, to a lower extent, on modelled PM2.5 due to the relevant role of NMVOC to the formation of fine secondary organic aerosols (SOA) (e.g. McDonald et al., 2018). For PM2.5, it is however important to note that the sensitivity of the modelling results to changes in the temporal profiles of NMVOC emissions will very much dependent on the SOA formation scheme that is implemented in the model. Currently, there are several models that include simplified SOA schemes, in which SOA precursor emissions from combustion sources are estimated using CO emissions as a proxy and therefore modelled SOA levels are not sensitive to changes in primary NMVOC emissions (e.g., Pai et al., 2020). Despite being a precursor of O3, we believe that changes in the temporal allocation of primary NMVOC emissions will have a rather low impact on O3 modelled concentrations. This hipothesis is based on results published by recent studies, which showed that significant changes in the amount or the speciation of anthropogenic NMVOC emissions translated into very limited or almost negligible changes of modelled O3 concentrations during the summer season, when O3 levels are at their maximum (Petetin et al., 2023; Oliveira et al., 2025).

We have included this discussion in the Conclusions section of the revised version of the manuscript.

McDonald, B.C., et al., Volatile chemical products emerging as largest petrochemical source of urban organic emissions, Science, 359,760-764, DOI:10.1126/science.aaq0524, 2018.

Petetin, H., Guevara, M., Garatachea, R., López, F., Oliveira, K., Enciso, S., Jorba, O., Querol, X., Massagué, J., Alastuey, A. and Pérez García-Pando, C.: Assessing ozone abatement scenarios in the framework of the Spanish ozone mitigation plan. Sci. Total Environ., 902 (2023), Article 165380, 10.1016/J.SCITOTENV.2023.165380.

Oliveira, K., Guevara, M., Kuenen, J., Jorba, O., Pérez García-Pando, C. and Denier van der Gon, H.: Enhancing anthropogenic NMVOC emission speciation for European air quality modelling, Environmental Pollution, 382, 126510, <a href="https://doi.org/10.1016/j.envpol.2025.126510">https://doi.org/10.1016/j.envpol.2025.126510</a>, 2025.

Pai, S. J., Heald, C. L., Pierce, J. R., Farina, S. C., Marais, E. A., Jimenez, J. L., Campuzano-Jost, P., Nault, B. A., Middlebrook, A. M., Coe, H., Shilling, J. E., Bahreini, R., Dingle, J. H., and Vu, K.: An evaluation of global organic aerosol schemes using airborne observations, Atmos. Chem. Phys., 20, 2637–2665, https://doi.org/10.5194/acp-20-2637-2020, 2020.

**Minor issues identified:**

Line 68: A verb appears to be missing—perhaps "show"?

Yes, the verb show was missing. We added it.

Line 241: The text refers to "GENEMIS-Menuetal2012" and later "Menutetal"—this should be made consistent or corrected.

Thanks for spotting this error. We have replaced "GENEMIS-Menuetal2012" by "GENEMIS-Menutetal2012"

Line 458: It would be helpful to briefly clarify that the "Harmut cold spell" refers to a negative temperature anomaly, explicitly.

We clarified this point as follows:

"These profiles lead to an increase of the total NOx emissions during February when compared to TNO (10%) and GENEMIS (2%), as shown in Fig. S1, reflecting the Hartmut cold spell, a winter storm that brought a cold wave and negative temperature anomalies to large areas of Europe during that month (C3S, 2018)." (lines 712 to 715 of the revised manuscript)

---

## Author Response (AR2)

**Reviewer #1**

**General comments**

The authors have adequately addressed all my previous questions and suggestions. I appreciate the effort put into the revisions. I just have a few remaining minor comments for consideration.

The authors would like to thank the reviewer for appreciating the effort put into the revisions. The answers to remaining minor comments can be found below in blue.

**Specific Comments**

Line 285: Should it be "GFASv2.1" or just "GFAS"? Both terms appear throughout the manuscript, but it would be helpful to use a single, consistent term to improve clarity.

We have replaced the term "GFAS" with "GFASv1.2" to be consistent throughout the manuscript.

Figures S4: Thank you for including Figure S4 as requested. However, this figure is not mentioned in the analysis presented in the Results section. Does it not provide any relevant insights? Even if that's the case, it would be good to briefly reference the figure and clarify its role. For example, when discussing Table 4 and 6, the authors could also refer to Figure S4 and comment on aspects such as: Where are the highest correlations found? Where are the lowest? Do the extremes occur in the same countries? What might explain these patterns?

Following the reviewer's suggestion, we have included multiple references to Figure S4 in the Results section:

"The monthly cycles obtained with the three temporal profile databases present correlations of 0.67 (CAMS-REG-TEMPO versus TNO) and 0.79 (CAMS-REG-TEMPO versus GENEMIS) (Table 4), with the highest correlations occurring in Finland (0.89 for CAMS-REG-TEMPO versus TNO) and Italy (0.87 for CAMS-REG-TEMPO versus GENEMIS) (Fig. S4)." (lines 268 to 270 of the revised manuscript)

"At the country level, maximum correlations occur in UK (0.9) and Czech Republic (0.91) when comparing CAMS-REG-TEMPO versus TNO and CAMS-REG-TEMPO versus GENEMIS, respectively. Negative correlations of -0.2 (CAMS-REG-TEMPO versus TNO) and -0.22 (CAMS-REG-TEMPO versus GENEMIS) are observed for Spain (Fig. S4), mainly due to the differences in the proposed profile for the agricultural waste burning emissions." (lines 291 to 293 of the revised manuscript)

"NH3 exhibits the largest differences in monthly emission distributions (Fig. 2), especially when comparing CAMS-REG-TEMPO and TNO profiles (correlation coefficient of 0.39, Table 4), the country-level monthly correlations showing large variations, with values

ranging from 0.88 (Malta) to -0.15 (Sweden) (Fig. S4)." (lines 304 to 306 of the revised manuscript)

"The GENEMIS profile is more in line with that of CAMS-REG-TEMPO (correlation coefficient of 0.78, Table 4, and 16 countries out of 29 showing correlations above 0.65, Fig. S4)," (lines 317 to 318 of the revised manuscript)

"Correlation values are generally consistent across individual countries, with 24 countries out of 29 presenting correlations above 0.8 (Fig. S4)." (lines 344 to 345 of the revised manuscript)

**Lines 355–362: The authors observe substantial differences in the correlations (Figures S4–S6) for countries with country-dependent profiles?**

As shown in Fig. S4 to S6, extreme (highest/lowest) correlation values are not always occurring in the same countries. Countries showing the largest/lowest correlation values vary depending on the pollutant considered. Country-level CAMS-REG-TEMPO versus TNO and CAMS-REG-TEMPO versus GENEMIS correlations depend on two factors: i) the contribution of each sector to overall emissions and ii) the differences between temporal profiles proposed by CAMS-REG-TEMPO, TNO and GENEMIS for the sectors that present the largest contributions to total emissions. Therefore, there is not a specific pattern in the resulting correlation coefficients that can be attributed to the use or not of country-dependent profiles. The reasons behind the extreme correlation values shown in Fig. S4 to S6 are related to multiple aspects that depend on the study case. Two examples are provided here to ilustrate this reasoning:

Spain: Monthly correlations for PM2.5 are 0.88 (CAMS-REG-TEMPO versus TNO) and 0.94 (CAMS-REG-TEMPO versus GENEMIS), while for NMVOC the correlations are -0.2 (CAMS-REG-TEMPO versus TNO) and -0.22 (CAMS-REG-TEMPO versus GENEMIS). The lower correlations reported for NMVOC when compared to PM2.5 are due to: i) the larger contribution of agricultural waste burning emissions (GNFR\_L) to total NMVOC emissions when compared to total PM2.5 emissions (15.6% versus 2.6%) and ii) the large differences between the CAMS-REG-TEMPO, TNO and GENEMIS monthly profiles proposed for agricultural waste burning emissions. Correlations for PM2.5 monthly emissions are large because more than 55% of total PM2.5 emissions are related to the use of residential and commercial combustion sector (GNFR\_C), for which CAMS-REG-TEMPO, GENEMIS and TNO propose similar monthly profiles, with emissions increasing during cold months.

Figure - NMVOC emission temporal distributions obtained per sector for Spain when using the CAMS-REG-TEMPO, TNO and GENEMIS profiles, respectively

Figure –  $PM_{2.5}$  emission temporal distributions obtained per sector for Spain when using the CAMS-REG-TEMPO, TNO and GENEMIS profiles, respectively

Cyprus: Monthly correlations for NMVOC are 0.87 (CAMS-REG-TEMPO versus TNO) and 0.79 (CAMS-REG-TEMPO versus GENEMIS), while for NO $_{\rm x}$  the correlations are -0.42 (CAMS-REG-TEMPO versus TNO) and -0.71 (CAMS-REG-TEMPO versus GENEMIS). The lower correlations reported for NO $_{\rm x}$  when compared to NMVOC are due to: i) the larger contribution of energy industry emissions (GNFR\_A+B) to total NO $_{\rm x}$  emissions when compared to total NMVOC emissions (45.7 % versus 2.8%) and ii) the large differences between the CAMS-REG-TEMPO, TNO and GENEMIS monthly profiles proposed for the

energy industry sector. Correlations for NMVOC monthly emissions are large because more than 60% of total NMVOC emissions are related to the use of solvent sector (GNFR\_E), for which CAMS-REG-TEMPO, GENEMIS and TNO propose the same monthly profile.

Figure –  $NO_x$  emission temporal distributions obtained per sector for Cyprus when using the CAMS-REG-TEMPO, TNO and GENEMIS profiles, respectively

Figure – NMVOC emission temporal distributions obtained per sector for Cyprus when using the CAMS-REG-TEMPO, TNO and GENEMIS profiles, respectively

Lines 393–395: Please verify the reported correlations. They seem to be reversed. Shouldn't it be 0.94 for CAMS-REG-TEMPO and GENEMIS, and 0.89 for CAMS-REG-TEMPO and TNO?

The reviewer is right. The correlation values were reversed. We corrected it.

Figure 2: I suggest not numbering this as Figure 2. Instead, it could be labeled as Figure 1 with the caption "(continued)". Additionally, the figure is not referenced in the analysis. The same comment applies to Figures 4 and 5.

We decided to keep the original labelling for Figures 2, 4 and 6, following with what is proposed in other Copernicus publications (e.g., Guevara et al., 2023; https://essd.copernicus.org/articles/14/2521/2022/).

Figures 2, 4 and 5 are referenced at the beginning of Section 3.1, as well as in different parts of the text, for instance:

"Figure 1 to Figure 6 compare the monthly, weekly and hourly emission temporal distributions for key pollutants" (line 252 of the revised manuscript)

"NH3 exhibits the largest differences in monthly emission distributions (Fig. 2), especially" (line 304 of the revised manuscript)

"For SOx (Fig. 5), differences in hourly emission cycles are rather small" (line 461 of the revised manuscript)

We have added a reference to Fig. 2 that was missing:

"For PM10 (Fig. 2), all three temporal profile datasets allocate" (line 324 of the revised manuscript)

We also found two cases where we were wrongly referencing to Fig.3 instead of Fig.4. We have corrected these two cases as follows:

"(···) with all three datasets assuming a near-flat weekly distribution of emissions (Fig. 4)." (lines 397 to 398 of the revised manuscript)

"For PM10 and PM2.5, similar discrepancies are observed across datasets (Fig. 4)." (line 400 of the revised manuscript)

Section 3: The manuscript states that the analysis was conducted by season, which is not entirely accurate. As mentioned in the first round of review, in Europe, the winter months are December, January, and February. The authors have actually conducted a quarterly (three-month) analysis, not a seasonal one.

We agree with the reviewer. We have replaced the term "season" by "quarter" both in the revised versions of the manuscript and the Supplementary Material.

Line 678: Please consider replacing "7 models" with "7 out of 11 models" for greater precision.

**Replaced**

Figure 10: It's not quite accurate to say that JFM corresponds to winter or that JAS corresponds to summer, as these periods span across different meteorological

seasons. In Europe, winter runs from December to February and summer from June to August. What are the highest values referring to emissions? Clarifying this in the figure captions would be helpful for the reader. In general, it might be clearer if each figure were introduced and explained at the beginning of the corresponding subsection (as the authors did in Section 3.3), rather than relying heavily on long figure captions. For example, in Figure 10, the caption says: "The JFM and JAS periods were selected because they represent winter-like and summer-like conditions as well as the highest and lowest values of the year."

We have replaced the concepts "winter-like" and "summer-like" conditions by "cold weather" and "hot weather" conditions. The highest values refer to observed concentrations, not emissions. This aspect has also been clarified.

Following the reviewer's comment, we have removed these explanations from Figure 10 caption and added them in a new paragraph that introduces all figures related to NO2 results (beginning of Section 3.2.1):

"Figures 9 to 11 show the comparison between the observed and modelled  $NO_2$  monthly, weekly and diurnal cycles for the ENS and the spatial median of the monthly, weekly and diurnal temporal correlations obtained for the ENS and each individual CAMS model in expA and expB. For the weekly (Fig. 10) and diurnal (Fig. 11) results, selected quarters are shown because they represent cold and hot weather conditions, periods with the highest and lowest observed  $NO_2$  concentrations of the year or periods were the ENS show an improvement and deterioration of the correlation when using CAMS-REG-TEMPO, respectively. Results for the remaining quarters are reported in the Supplementary material (Figure S8 and S9)." (lines 574 to 579 of the revised manuscript)

We have followed the same approach in Sections 3.2.2 ( $O_3$  results), 3.2.3 ( $PM_{10}$  results) and 3.2.4 ( $PM_{2.5}$  results).